# Effectiveness and safety of immunosuppressive regimens used as maintenance therapy in kidney transplantation: The CESIT study

Arianna Bellini [1,2], Marco Finocchietti[1], Alessandro Cesare Rosa[1], Maurizio Nordio[3], Eliana Ferroni[3], Marco Massari[4], Stefania Spila Alegiani[4], Lucia Masiero[5], Gaia Bedeschi[5], Massimo Cardillo[5], Ersilia Lucenteforte[6], Giuseppe Piccolo[7], Olivia Leoni[8], Silvia Pierobon[3], Stefano Ledda[9], Donatella Garau[9], Marina Davoli[1], Antonio Addis[1], Valeria Belleudi [1] *, on behalf of CESIT study group[¶]

1 Department of Epidemiology, Lazio Regional Health Service, Rome, Italy, 2 Department of Public Health and Infectious Diseases, Sapienza University of Rome, Rome, Italy, 3 Azienda Zero of the Veneto Region, Padua, Italy, 4 National Centre for Drug Research and Evaluation, Istituto Superiore Di Sanità, Rome, Italy, 5 Italian National Transplant Centre, Istituto Superiore di Sanità, Rome, Italy, 6 Department of Clinical and Experimental Medicine, University of Pisa, Pisa, Italy, 7 Regional Transplant Coordination, Lombardy Region, Milan, Italy, 8 Department of Health of Lombardy Region, Epidemiology Observatory, Milan, Italy, 9 General Directorate for Health, Sardinia Region, Italy

☯ These authors contributed equally to this work.
¶ Membership of the CESIT study group is provided in the acknowledgments
* v.belleudi@deplazio.it

**Data Availability Statement:** The data that support the findings of this study are available from the Italian regions participating to CESIT study but

## Abstract

Maintenance immunosuppressive therapy used in kidney transplantation typically involves calcineurin inhibitors, such as tacrolimus or cyclosporine, in combination with mycophenolate or mechanistic target of rapamycin (mTORi) with or without corticosteroids. An Italian retrospective multicentre observational study was conducted to investigate the risk-benefit profile of different immunosuppressive regimens. We identified all subjects who underwent kidney transplant between 2009 and 2019, using healthcare claims data. Patients on cyclosporine and tacrolimus-based therapies were matched 1:1 based on propensity score, and effectiveness and safety outcomes were compared using Cox models (HR; 95%CI). Analyses were also conducted comparing mTORi versus mycophenolate among tacrolimus-treated patients. Patients treated with cyclosporine had a higher risk of rejection or graft loss (HR:1.69; 95%CI:1.16–2.46) and a higher incidence of severe infections (1.25;1.00–1.55), but a lower risk of diabetes (0.66;0.47–0.91) compared to those treated with tacrolimus. Among tacrolimus users, mTORi showed non-inferiority to MMF in terms of mortality (1.01;0.68–1.62), reject/graft loss (0.61;0.36–1.04) and severe infections (0.76;0.56–1.03). In a real-life setting, tacrolimus-based immunosuppressive therapy appeared to be superior to cyclosporine in reducing rejection and severe infections, albeit with an associated increased risk of diabetes. The combination of tacrolimus and mTORi may represent a valid alternative to the combination with mycophenolate, although further studies are needed to confirm this finding.

restrictions apply to the availability of these data, which were used under license (as by third-party sources) for the current study, and so are not publicly available. However, data are available with permission of Italian regions, which are the data owner. The non-author contact information to which data requests may be sent is: project. cesit@gmail.com.

**Funding:** This work was supported by the Italian Medicines Agency in the context of the multiregional pharmacovigilance project (AIFA 2012–2014: Comparative Effectiveness and Safety of Immunosuppressive Drugs in Transplant patients—CESIT project). Grant code: J85I2000009005 (CUP). The funders had no role in study design, data collection and analysis, decision to publish, or preparation of the manuscript.

**Competing interests:** The authors have declared that no competing interests exist.

## Introduction

Kidney transplantation is the preferred form of renal replacement therapy for the majority of patients with end-stage renal disease, offering known clinical and economic benefits over dialysis [1]. The central challenge in organ transplantation remains the suppression of allograft rejection; thus, immunosuppressive drugs play a pivotal role in achieving successful allograft function [2].

KDIGO Guidelines [3] recommend a maintenance immunosuppressive therapy that includes a combination of calcineurin inhibitor-CNI (cyclosporine-CsA or tacrolimus-TAC) and an antiproliferative agent (azathioprine or mycophenolate-MMF), with or without corticosteroids. Then, the mammalian target of rapamycin inhibitors-mTORi (everolimus or sirolimus), introduced in the Italian market recently, may be considered in combination with CNI, with or without corticosteroids as an alternative to antimetabolites. The National Institute for Health and Care Excellence (NICE) guidelines [4], published in 2017, identified TAC, MMF, and mTORi as possible options for maintenance therapy, emphasizing that limited conclusions can be drawn regarding clinical effectiveness differences among these options. To date, no consensus has emerged on the optimal drug combination in terms of safety and effectiveness for renal recipients; the most commonly prescribed therapy both in USA [5] and South-Eastern Europe [6] is a triple regimen with TAC, MMF, and corticosteroids.

In the past years, many studies have compared different agents, primarily within the same therapeutic category: CNIs, considered the cornerstone of immunosuppressive therapy post-kidney transplant, have been evaluated in several randomized controlled trials (RCTs) [7,8], with TAC emerging as a superior therapy for improving graft survival and reducing acute rejection. Furthermore, there might be advantages to using MMF over azathioprine, both in combination with TAC or CsA, for the prevention of rejection [9,10]; however, there are still some controversies in terms of efficacy and safety when considering MMF and mTORi [11–13].

As such, previous evidence show how recommendations for immunosuppressive regimens are complex, due to the combination of multiple classes of drugs, and that the choice among different strategies involves trade-off between benefits and risks, considering various factors for both patients and donors.

In this context, there are limited data from observational studies analysing immunosuppressive strategies used in European countries after renal transplantation in clinical practice. Randomized Controlled Trials (RCTs) may not fully reflect real-world clinical practice for transplant patients, as the trial population, selected on the base of restrictive enrollment criteria, may not represent the broader population that will use these drugs and may not include patients with a wider range of ages and varying comorbidities. Moreover, real-world evidence may detect outcomes that require long-term follow-up (such as mortality, cancer, infection) and may highlight specific factors, not evident in RCTs, that can influence treatment choices and clinical outcomes (such as drug costs, adherence, switching).

This work was conducted within the context of the multiregional active pharmacovigilance CESIT project with the aim of improving knowledge about maintenance immunosuppressive therapies prescribed after solid organ transplantation [14]. Recently, the study group has published an article focused on the immunosuppressive drug utilization patterns among Italian patients who underwent kidney transplantation, showing that a considerable variability in dispensation patterns exists across years, regions, and centres in the country [15].

Along these lines the present work aims to compare the effectiveness and safety profile of the different immunosuppressive therapeutic regimens prescribed for renal recipients in four Italian regions between 2009 and 2019.

## Methods

This study was approved by the Ethical Committee of the Local Health Authority Roma 1, the reference ethic committee of the Department of Epidemiology of Lazio (the CESIT coordinating centre), according to the current national law. Informed consent was obtained from each patient and/or the legally acceptable representative (LAR). The study was conducted in accordance to relevant guidelines and regulations.

The study is a retrospective multicentre observational cohort study, involving four Italian regions (Lombardy, Veneto, Lazio, Sardinia, covering a total population of over 20 million inhabitants) and based on data from regional healthcare claims and the national transplant information system (data were accessed in December 2021). National transplant information system is an infrastructure for the management of data related to the activity of the National Transplant Network, established and regulated by Italian Laws (n. 91 of April 1, 1999).

Specifically, regional analytical datasets pertaining to incident patients who underwent kidney transplants in the years 2009–19 were created with information extracted, through a common data model, from hospital information, pharmaceutical dispensation, mortality information systems and co-payment exemption registry. This was facilitated by a distributed analysis tool called The ShinISS [16].

Information on demographical and clinical characteristics of donor and receipt, available nationwide, was linked through a semi-deterministic matching. Details on this procedure are described elsewhere [14,15].

The study cohort was restricted to patients with no previous single or multi-organ transplantations; residing in the regions considered, surviving and with at least one CNI immunosuppressive dispensation during the 30 days post discharge (index period).

Patients were categorized based on the calcineurin inhibitor used during the index period in: either TAC or CsA. Among patients in the TAC group, a further distinction was made between MMF or mTORi combination. Patients under azathioprine treatment were excluded.

Each patient, starting from 30 days post discharge, was tracked until the occurrence of the study event (i.e., death) or the end of the study, for a maximum of five years, whichever came first. The considered outcomes were, mortality and transplant rejection/graft failure for effectiveness analysis, and the incidence of severe infections, cancer, diabetes, major adverse cardiovascular events (MACE) and statin use for safety analysis. Data on transplant rejection recorded in the national transplant information system was directly reported by clinicians upon histologically documented immunological cause leading to functional impairments of the transplanted organ.

The infection-related outcome focused solely on severe infections, defined as those necessitating hospitalization. The selected ICDIX-CM codes for this outcome are provided in the S1 Table, the choice of codes was based on some previous work published in the literature identifying the most relevant infections in the post-transplant population [17–21].

For the main analysis was employed the intention-to-treat (ITT) approach. Patients receiving CsA- and TAC-based therapies during the index period, were matched 1:1 by propensity score (PS) nearest neighbor approach without replacement, with a caliper of 0.1 [22]. PS-matching was established considering region of residence, demographic characteristics of the donor and recipient (sex and age), type of donor, information on transplant (indication, dialysis history, panel reactivity antibodies, number of total and specific mismatch (Human leukocyte antigens-HLA-A, HLA-B, HLA-DR)), length of transplant hospitalization (prolonged hospitalization was defined as a length of stay equal to or greater than the 75th percentile of length of stay of all participants), year of discharge, clinical history in terms of comorbidity (hypertension, diabetes, cardio-cerebrovascular diseases, cancer, hematologic

diseases, thyroid disorders) and comedication (anticoagulants, antianemics, antiplatelet, diuretics, statins). To assess covariate balance after PS-matching, the standardized mean difference (SMD) between groups, CsA or TAC users, was calculated.

In the risk-effectiveness analysis, only patients who were at risk of developing the outcome for the first time were considered; for each specific outcomes, patients with a prior history of the considered event were excluded. Treatment effectiveness and safety were estimated comparing outcomes between groups using a Cox model (hazard ratio-HR; 95%CI). Analyses were replicated by comparing mTORi vs MMF within patients in TAC therapy. Kaplan-Meier (KM) curves were presented and the cumulative risk was compared between groups using log-rank test.

To ensure the consistency of our results, an as-treated (AT) approach was applied. This involved censoring patients who interrupted treatment (by not refilling a prescription within 90 days after the expiration of the last prescription's supply) or had a switch in immunosuppressive treatment during follow-up (e.g., patients in the TAC-based therapy group receiving a CsA prescription during follow-up were censored at the date of dispensation, and vice versa). The same procedure was applied for the mTORi vs MMF comparison.

Moreover, subgroup and sensitivity analyses were conducted. First, the effectiveness and safety profile was calculated after stratifying the cohort according to age class (18–29 years; 30–59 years; 60+ years). Secondly, HRs were calculated by adjusting for prednisone use, in order to eliminate potential disproportionality in the use of steroids in the two comparison groups. The potential role of previous infections and tumours in the donor was also examined. The cohort was restricted to years where this information was available, and the association between immunosuppressive regimen and outcomes was calculated adjusting for them. In fact, transplantation carries an unavoidable risk of transmission of malignant diseases, which may be heightened when the organ is from donors with a history or ongoing malignancy [23]. Additionally, diverse donor-infections, particularly viral infections including Cytomegalovirus (CMV), have been recognized in transplant recipients [24]. Finally, since delayed graft function (DGF) is a major obstacle for allograft survival, the primary analysis was also re-run after adjusting the model for DGF.

All analyses were performed using SAS Statistical Software version 9.4 (SAS Institute Inc., Cary, NC) and Statistical package R version 4.1.3.

## Results

Overall, 5,318 residents who underwent kidney transplantation discharge in the four Italian regions under study during the period 2009–2019 were selected. After applying the selection criteria, 3,622 (68.1%) kidney recipients were considered, of which 787 (21.7%) were treated with CsA-based therapy and 2,835 (78.3%) with TAC-based therapy. Among patients in TAC-based therapy, 69.9% were in combination with MMF and 19.7% with mTORi (everolimus 90.2%–sirolimus 9.8%). Patients in triple therapy, CNI+MMF+Pred, were 416 in CsA group and 1,682 in the TAC group (Fig 1).

Although before PS-matching, considerable differences in several variables (e.g., region, type of donor, *Panel reactive antibody* (PRA), total number of HLA mismatches) were found between the comparison groups (Figs 2 and 3), after matching, the distribution of all baseline characteristics was well balanced with standardized difference ≤ 0.1 (Table 1). Following PS-matching, a total of 1,438 and 890 patients were included in the CsA-TAC and TAC+mTORi–TAC+MMF matched cohort, respectively. Notably, although only total mismatch was included in the PS-matching, specific mismatch data (HLA-A, HLA-B, HLA-DR) were also reported in the table and were found to be comparable among the groups.

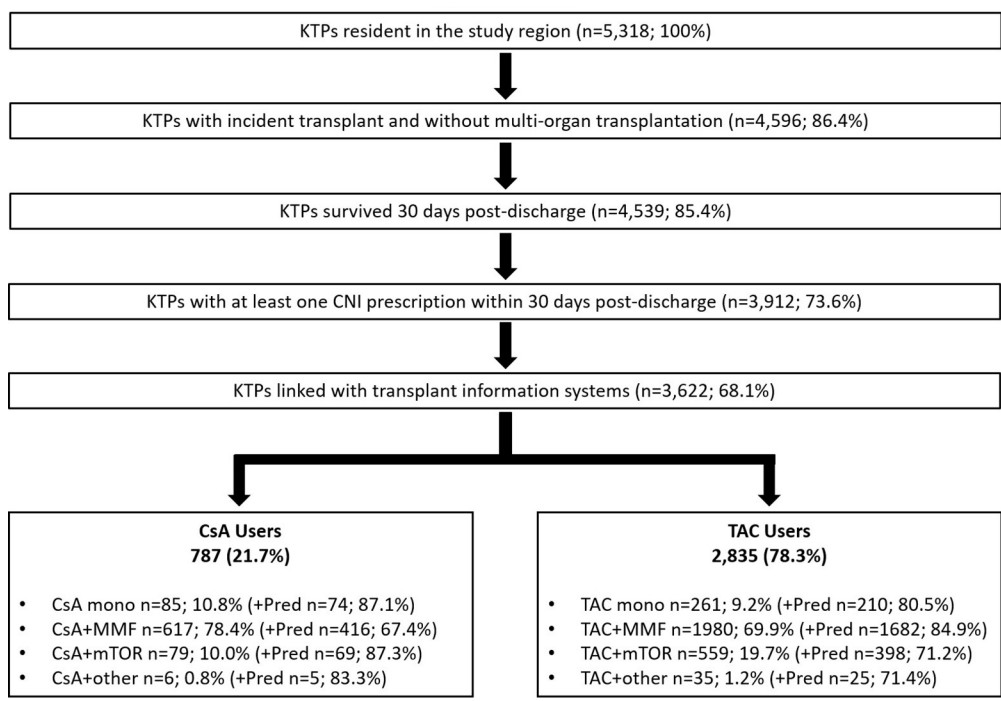

**Fig 1. Flow chart of subject inclusion and exclusion criteria.**

After matching, the median follow-up time was 4.2 years (1.9–5.0) for the cohort of TAC and CsA users and 3.4 years (1.6–5.0) for the cohort of TAC+MMF and TAC+mTORi users.

In the first comparison, HRs were estimated for study outcomes among individuals using CsA- versus those using TAC- based-therapy (Fig 4). Patients treated with CsA had higher risk of rejection/graft loss (HR:1.46; 95%CI 1.02–2.09) and an incidence of severe infections (HR1.28; 95%CI 1.01–1.61), and a lower risk of new-onset diabetes (HR:0.71; 95%CI 0.51–1.00) compared to those treated with TAC.

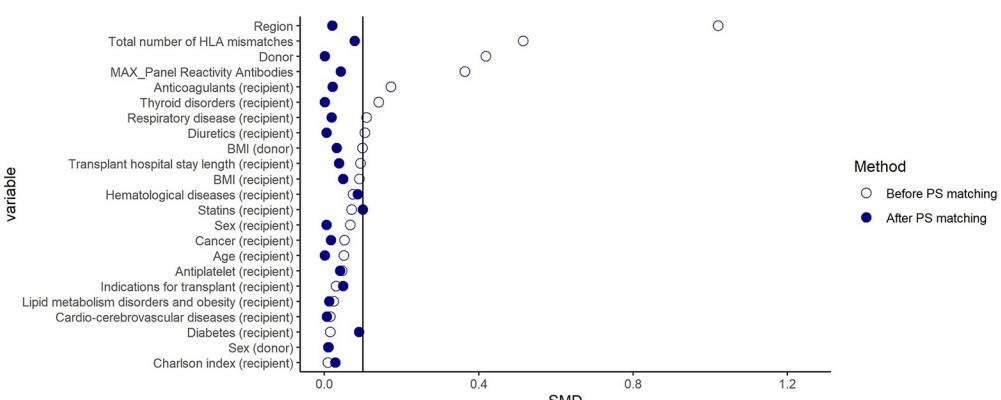

**Fig 2. Plot of standardized mean differences among TAC and CsA users before and after propensity score (PS) matching. Note.** CsA: Cyclosporine; TAC: Tacrolimus; mTORi: Mammalian target of rapamycin inhibitors; MMF: Mycophenolate; IR: Incidence Rate; PY: Person-years; HR: Hazard ratio; 95%CI: 95% confidence interval; MACE: Major adverse cardiovascular events.

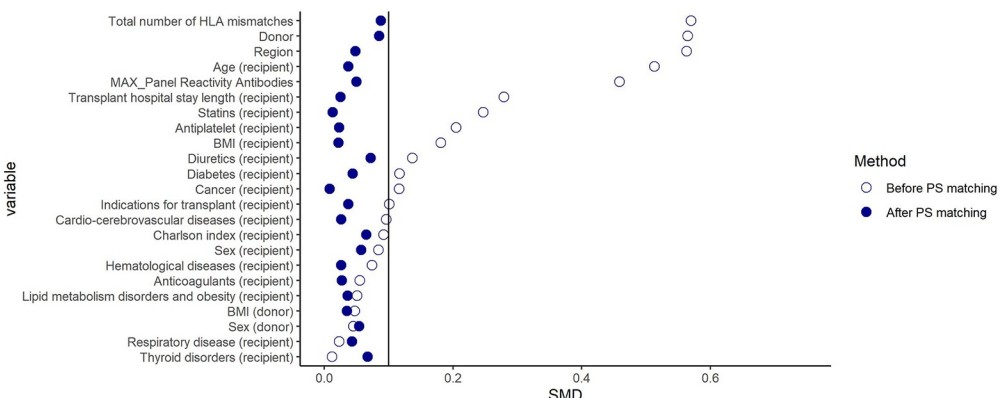

**Fig 3. Plot of standardized mean differences among mTORi and MMF users within patients treated with TAC before and after propensity score (PS) matching. Note.** CsA: Cyclosporine; TAC: Tacrolimus; mTORi: Mammalian target of rapamycin inhibitors; MMF: Mycophenolate; IR: Incidence Rate; PY: Person-years; HR: Hazard ratio; 95%CI: 95% confidence interval; MACE: Major adverse cardiovascular events.

While, among TAC users, patients assuming mTORi had a higher risk of incident use of statins (HR:1.61; 95%CI 1.19–2.19) compared to those assuming MMF (Fig 5). Furthermore, there was a trend towards a reduced risk of rejection/graft loss (HR:0.61; 95%CI 0.36–1.04) and severe infections (HR:0.76; 95%CI:0.56–1.03) was noted in the mTORi group even if the result did not reach statistical significance.

The KM curves comparing the cumulative incidence of considered outcomes in the two groups were consistent with these findings (Figs 6–12 and 13–19A–19G).

The rejection/graft loss outcome curves (Fig 7) initially overlapped within the first year of observation and then separated with a progressive increase in the distance between the two, with TAC-users demonstrating a lower risk of rejection/graft loss. Similarly, Fig 8 shows that the risk of severe infections between TAC and CsA users during the first year of follow-up was comparable, while the cumulative risk in the following years was higher for CsA users. Fig 9 illustrates that within the first year of follow-up, there was a very early separation of the KM curves that remained relatively constant for the rest of the observation period. When considering mTORi+TAC and MMF+TAC groups, the KM curves referring to the mortality outcome showed better survival for TAC+mTORi users in the first three years, although this did not

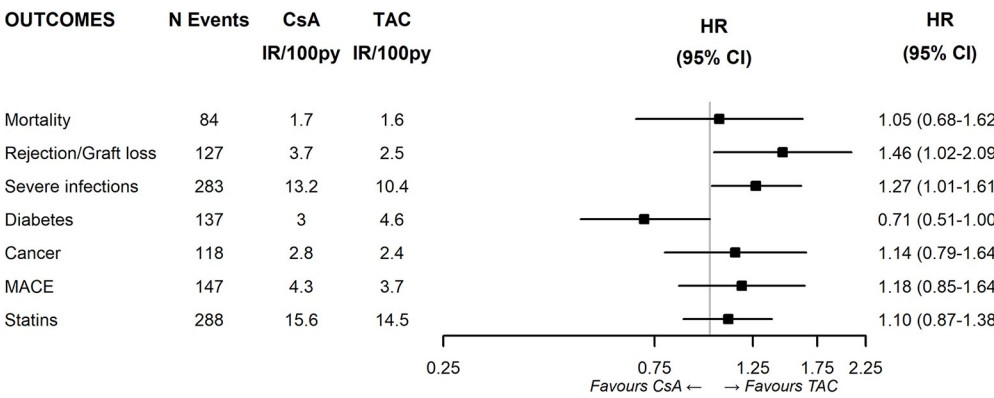

| OUTCOMES | N Events | CsA IR/100py | TAC IR/100py | HR (95% CI) | HR (95% CI) |
|---|---|---|---|---|---|
| Mortality | 84 | 1.7 | 1.6 | | 1.05 (0.68-1.62) |
| Rejection/Graft loss | 127 | 3.7 | 2.5 | | 1.46 (1.02-2.09) |
| Severe infections | 283 | 13.2 | 10.4 | | 1.27 (1.01-1.61) |
| Diabetes | 137 | 3 | 4.6 | | 0.71 (0.51-1.00) |
| Cancer | 118 | 2.8 | 2.4 | | 1.14 (0.79-1.64) |
| MACE | 147 | 4.3 | 3.7 | | 1.18 (0.85-1.64) |
| Statins | 288 | 15.6 | 14.5 | | 1.10 (0.87-1.38) |

**Fig 4. Effectiveness and safety of CsA vs TAC.**

**Table 1. Baseline covariate and standardized mean difference (SMD) post Propensity Score (PS) comparison pairs: TAC vs CsA and mTORi vs MMF combined with TAC-based therapy.**

| | TAC | | CsA | | SMD | TAC+MMF | | TAC+mTORi | | SMD |
|---|---|---|---|---|---|---|---|---|---|---|
| | **723** | | **723** | | | **446** | | **446** | | |
| | **n** | **%** | **n** | **%** | | **n** | **%** | **n** | **%** | |
| **Region** | | | | | | | | | | |
| Veneto | 85 | 11.8% | 87 | 12.0% | 0.021 | 174 | 39.0% | 164 | 36.8% | 0.048 |
| Lombardy | 563 | 77.9% | 557 | 77.0% | | 130 | 29.1% | 136 | 30.5% | |
| Latium | 75 | 10.4% | 79 | 10.9% | | 108 | 24.2% | 110 | 24.7% | |
| Sardinia | - | 0.0% | - | 0.0% | | 34 | 7.6% | 36 | 8.1% | |
| **Sex (recipient)** | | | | | | | | | | |
| Female | 237 | 32.8% | 239 | 33.1% | 0.006 | 140 | 31.4% | 152 | 34.1% | 0.057 |
| **Age (recipient)** | | | | | | | | | | |
| mean | 52.2 | | 52.3 | | 0.001 | 56.4 | | 56.9 | | 0.037 |
| median (1° quartile-3° quarile) | 54 | (45–63) | 55 | (45–63) | | 58 | 50–65 | 59 | 50–66 | |
| **BMI (recipient)** | | | | | | | | | | |
| underweight | 63 | 8.71% | 55 | 7.6% | 0.049 | 18 | 4.0% | 17 | 3.8% | 0.022 |
| normal range | 358 | 49.5% | 371 | 51.3% | | 240 | 53.8% | 244 | 54.7% | |
| overweight | 243 | 33.6% | 237 | 32.8% | | 145 | 32.5% | 144 | 32.3% | |
| obese | 59 | 8.2% | 60 | 8.3% | | 43 | 9.6% | 41 | 9.2% | |
| **Transplant hospital stay lenght** | | | | | | | | | | |
| standard hospitalization (≤ 18 days) | 529 | 73.2% | 541 | 74.8% | 0.038 | 375 | 84.1% | 379 | 85.0% | 0.025 |
| prolonged hospitalization (>19 days) | 194 | 26.8% | 182 | 25.2% | | 71 | 15.9% | 67 | 15.0% | |
| **Indications for transplant** | | | | | | | | | | |
| Cystic nephropathies | 132 | 18.3% | 135 | 18.7% | 0.049 | 97 | 21.7% | 99 | 22.2% | 0.037 |
| Glomerular nephropathies | 321 | 44.4% | 304 | 42.0% | | 189 | 42.4% | 181 | 40.6% | |
| other | 270 | 37.3% | 284 | 39.3% | | 160 | 35.9% | 166 | 37.2% | |
| **Donor** | | | | | | | | | | |
| Deceased | 705 | 97.5% | 705 | 97.5% | <0.001 | 443 | 99.3% | 439 | 98.4% | 0.085 |
| Living | 18 | 2.5% | 18 | 2.5% | | 3 | 0.7% | 7 | 1.6% | |
| **Sex (donor)** | | | | | | | | | | |
| Female | 331 | 45.8% | 335 | 46.3% | 0.011 | 200 | 44.8% | 212 | 47.5% | 0.054 |
| **BMI (donor)** | | | | | | | | | | |
| underweight | 29 | 4.0% | 25 | 3.5% | 0.032 | 11 | 2.5% | 12 | 2.7% | 0.035 |
| normal range | 347 | 48.1% | 348 | 48.3% | | 182 | 40.9% | 188 | 42.2% | |
| overweight | 247 | 34.3% | 252 | 35.0% | | 176 | 39.6% | 173 | 38.9% | |
| obese | 98 | 13.6% | 96 | 13.3% | | 76 | 17.1% | 72 | 16.2% | |
| missing | 2 | | 2 | | | 1 | | 1 | | |
| **Transplant characteristics** | | | | | | | | | | |
| **Panel Reactivity Antibodies (PRA)** | | | | | | | | | | |
| 0–20 | 661 | 91.9% | 662 | 91.8% | 0.043 | 410 | 92.1% | 414 | 93.0% | 0.05 |
| 21–79 | 37 | 5.1% | 38 | 5.3% | | 29 | 6.5% | 24 | 5.4% | |
| 80+ | 21 | 2.9% | 21 | 2.9% | | 6 | 1.3% | 7 | 1.6% | |
| *missing* | *4* | | *2* | | | *1* | | *1* | | |
| **Total number of HLA mismatches** | | | | | | | | | | |
| 0 | 71 | 10.1% | 57 | 8.1% | 0.079 | 84 | 19.0% | 86 | 19.6% | 0.088 |
| 1–3 | 225 | 31.9% | 243 | 34.5% | | 135 | 30.5% | 135 | 30.8% | |
| 4–6 | 409 | 58.0% | 405 | 57.4% | | 224 | 50.6% | 218 | 49.7% | |
| **Number of HLA-A mismatches*** | | | | | | | | | | |

*(Continued)*

**Table 1.** (Continued)

| | | TAC | | CsA | | SMD | TAC+MMF | | TAC+mTORi | | SMD |
|---|---|---|---|---|---|---|---|---|---|---|---|
| | | **723** | | **723** | | | **446** | | **446** | | |
| | 0 | 156 | 22.1% | 152 | 21.6% | 0.014 | 131 | 29.6% | 139 | 31.7% | 0.097 |
| | 1–2 | 549 | 77.9% | 553 | 78.4% | | 312 | 70.4% | 300 | 68.3% | |
| **Number of HLA-B mismatches*** | | | | | | | | | | | |
| | 0 | 122 | 17.3% | 110 | 15.6% | 0.045 | 105 | 23.7% | 116 | 26.4% | 0.106 |
| | 1–2 | 583 | 82.7% | 595 | 84.4% | | 338 | 76.3% | 323 | 73.6% | |
| **Number of HLA-DR mismatches*** | | | | | | | | | | | |
| | 0 | 178 | 25.2% | 170 | 24.1% | 0.026 | 146 | 33.0% | 156 | 35.5% | 0.101 |
| | 1–2 | 527 | 74.8% | 535 | 75.9% | | 297 | 67.0% | 283 | 64.5% | |
| **Comorbidities and comedications** | | | | | | | | | | | |
| **Charlson index** | | | | | | | | | | | |
| | 0–1 | 587 | 81.2% | 590 | 81.6% | 0.029 | 361 | 80.9% | 367 | 82.3% | 0.065 |
| | 2 | 106 | 14.7% | 107 | 14.8% | | 70 | 15.7% | 61 | 13.7% | |
| | 3+ | 30 | 4.1% | 26 | 3.6% | | 15 | 3.4% | 18 | 4.0% | |
| **Cancer** | | 45 | 6.2% | 42 | 5.8% | 0.017 | 40 | 9.0% | 39 | 8.7% | 0.008 |
| **Diabetes** | | 158 | 21.9% | 132 | 18.3% | 0.09 | 100 | 22.4% | 92 | 20.6% | 0.044 |
| **Lipid metabolism disorders and obesity** | | 37 | 5.1% | 35 | 4.8% | 0.013 | 31 | 7.0% | 27 | 6.1% | 0.036 |
| **Thyroid disorders** | | 69 | 9.5% | 69 | 9.5% | <0.001 | 53 | 11.9% | 63 | 14.1% | 0.067 |
| **Hematological diseases** | | 82 | 11.3% | 103 | 14.2% | 0.087 | 60 | 13.5% | 64 | 14.3% | 0.026 |
| **Anaemia** | | 257 | 35.5% | 251 | 34.7% | 0.017 | 121 | 27.1% | 124 | 27.8% | 0.015 |
| **Cardio-cerebrovascular diseases** | | 156 | 21.6% | 158 | 21.9% | 0.007 | 107 | 24.0% | 112 | 25.1% | 0.026 |
| **Hypertension** | | 487 | 67.4% | 478 | 66.1% | 0.026 | 332 | 74.4% | 325 | 72.9% | 0.036 |
| **Respiratory disease** | | 67 | 9.3% | 63 | 8.7% | 0.019 | 47 | 10.5% | 53 | 11.9% | 0.043 |
| **Diuretics** | | 286 | 39.6% | 288 | 39.8% | 0.006 | 219 | 49.1% | 203 | 45.5% | 0.072 |
| **Anticoagulants** | | 49 | 6.8% | 53 | 7.3% | 0.022 | 54 | 12.1% | 58 | 13.0% | 0.027 |
| **Antiplatelet** | | 243 | 33.6% | 229 | 31.7% | 0.041 | 180 | 40.4% | 175 | 39.2% | 0.023 |
| **Epoetins** | | 215 | 29.7% | 199 | 27.5% | 0.049 | 172 | 38.6% | 197 | 44.2% | 0.114 |
| **Statins** | | 311 | 43.0% | 347 | 48.0% | 0.100 | 236 | 52.9% | 233 | 52.2% | 0.013 |

**Note**. TAC: Tacrolimus; CsA: Cyclosporine; mTORi: Mammalian target of rapamycin inhibitors; MMF: Mycophenolate SMD: Standardized Mean Differences.

*Variable not included in PS-matching procedure.

reach statistical significance, the curves reversed after the third year of observation (Fig 13). Fig 14 concerning rejection/graft loss shows that after about one year of observation, a separation between the two curves appeared, with a lower incidence of the outcome in TAC+mTORi users, and this difference was maintained throughout the follow-up period without reaching statistical significance. Similarly, Fig 15, focused on infections, reveals that the initial distance found between the two curves, with a lower occurrence of the outcome in TAC+mTORi users, decreased after the first year of observation and again did not reach significance. Also, KM curves for new-onset diabetes and statins use outcomes separated early, within the first six months of treatment initiation.

Different subgroup and sensitivity analyses were performed to verify the robustness of the observed results (Table 2).

Overall, results from the primary analysis did not change with the as-treated analytical approach; however, when considering the second comparison, the precision of the estimates was lower for rejection/graft loss (HR: 0.58; 95%CI 0.27–1.27) and severe infections (HR:0.82; 95%CI 0.58–1.16). In this context, it is interesting to report that among TAC and CsA users,

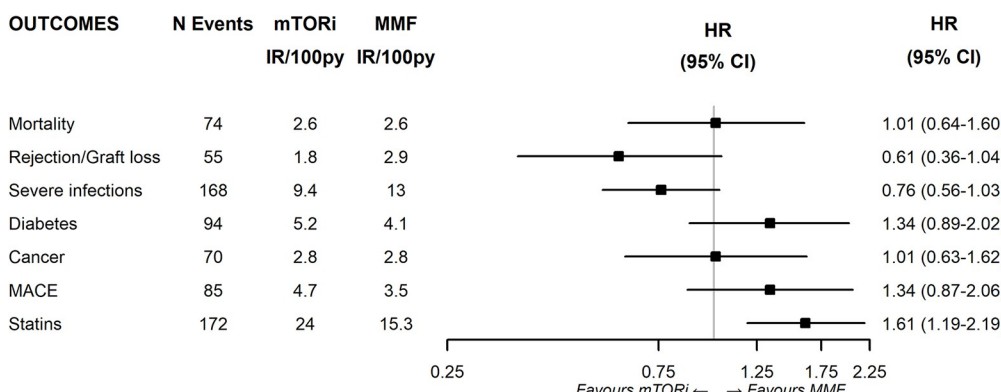

**Fig 5. Effectiveness and safety of mTORi vs MMF within patients treated with TAC. Note.** CsA: Cyclosporine; TAC: Tacrolimus; mTORi: Mammalian target of rapamycin inhibitors; MMF: Mycophenolate; PY: Person-years; HR: Hazard ratio; 95%CI: 95% confidence interval; MACE: Major adverse cardiovascular events.

the percentages of patients switching to the other CNIs were 5.7% and 9.3% respectively, while mTORi was replaced by MMF more frequently with respect to switching in the other direction (36.5% vs 9.9%) (Table 3).

When the cohort was stratified by age, the comparison between TAC and CsA did not show important differences from the main analysis, but statistical significance was not reached for reject/graft loss in the subgroup aged 60+ and for new-onset diabetes in the 18–29 years and 30–59 years groups for both outcomes all the HRs maintained their sign as in our original model.

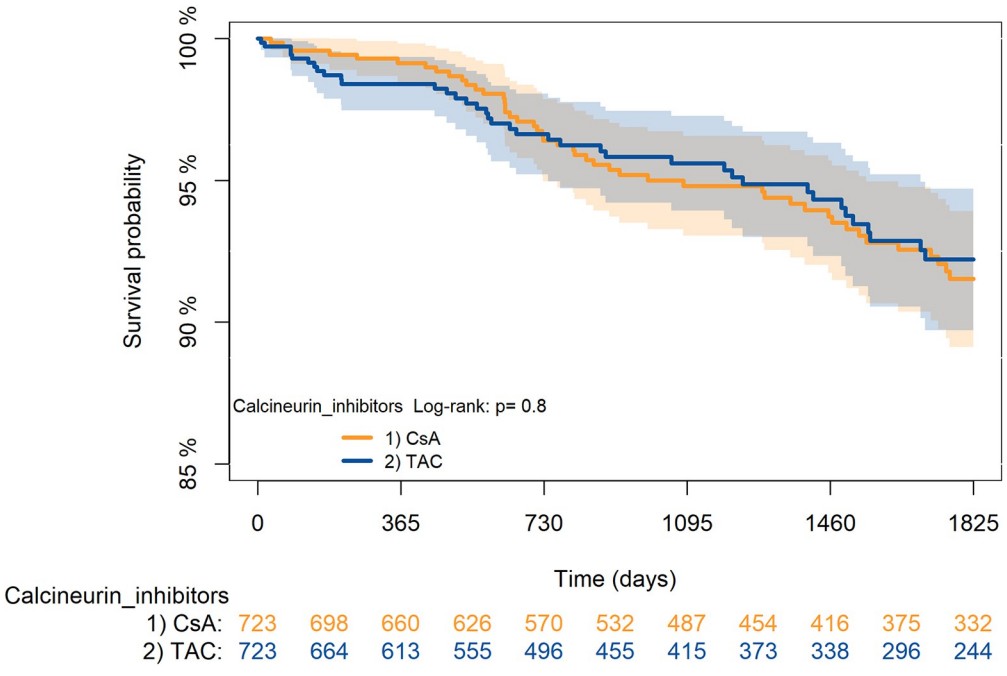

**Fig 6. Kaplan-Meier curves for survival according to TAC or CsA.**

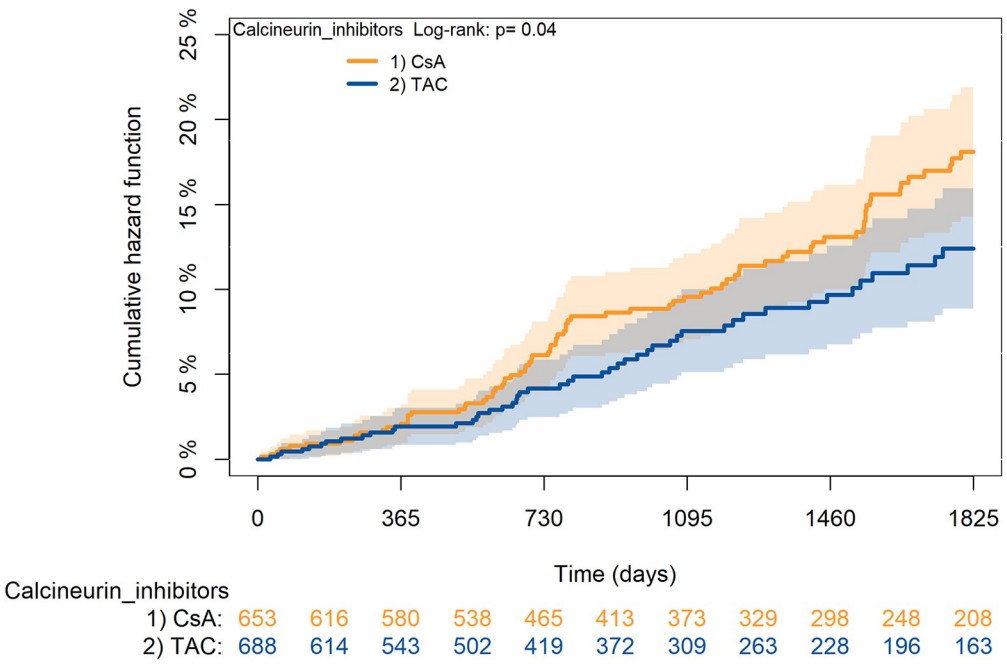

**Fig 7. Kaplan-Meier curves for rejection/graft loss according to TAC or CsA.**

When considering the comparison between mTORi and MMF, results showed that in the two youngest age groups mTORi users were at significantly lower risk of rejection/graft loss and at higher risk of using statins, while statistical significance was not reached for the oldest

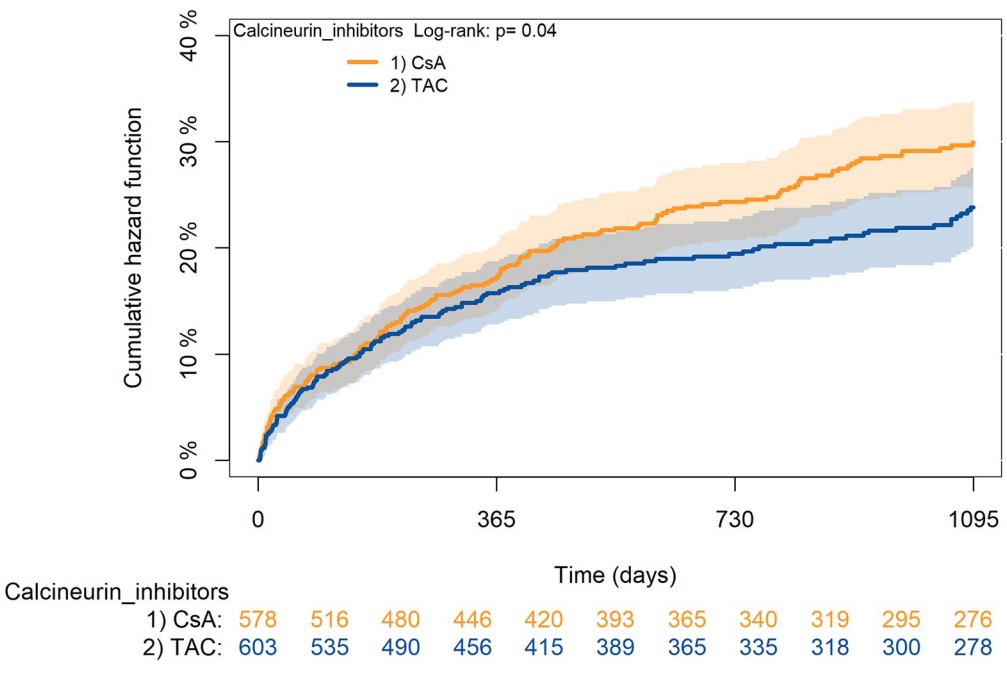

**Fig 8. Kaplan-Meier curves for severe infections according to TAC or CsA.**

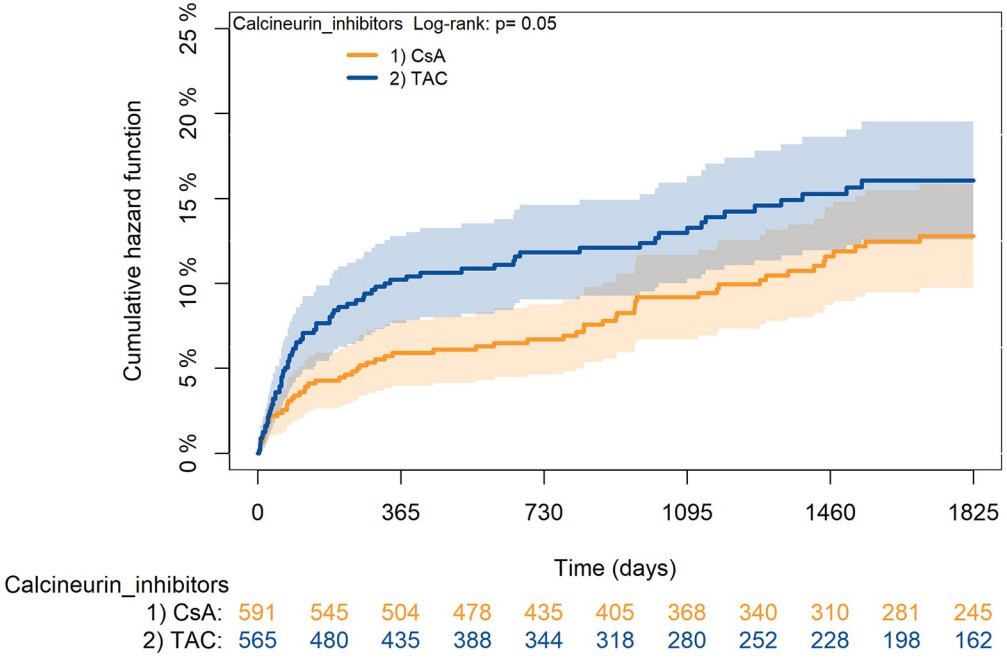

**Fig 9. Kaplan-Meier curves for diabetes according to TAC or CsA.**

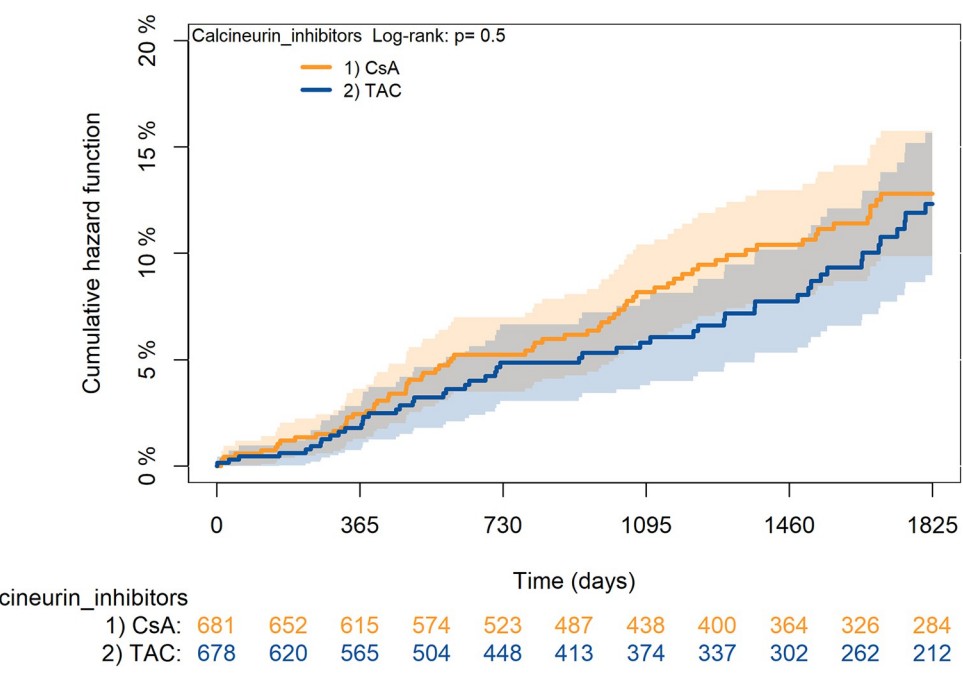

**Fig 10. Kaplan-Meier curves for cancer according to TAC or CsA.**

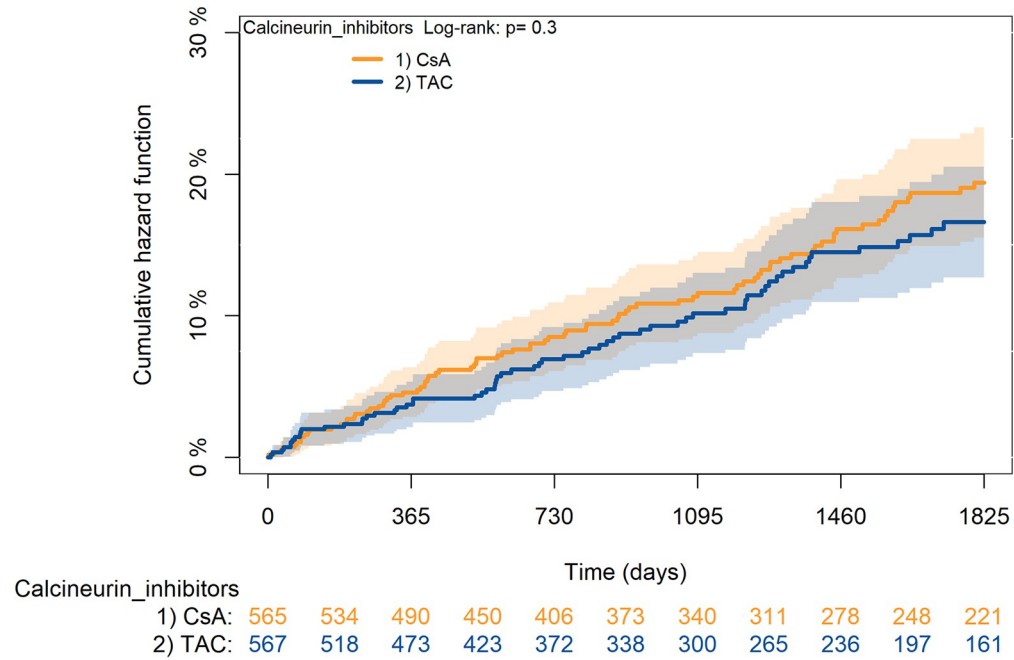

**Fig 11. Kaplan-Meier curves for MACA according to TAC or CsA.**

**Fig 12. Kaplan-Meier curves for statin use according to TAC or CsA.** A: Survival, B: Rejection/graft loss, C: Severe infections, D: Diabetes, E: Cancer, F: MACE, G: Statin use. Note. CsA: Cyclosporine; TAC: Tacrolimus.

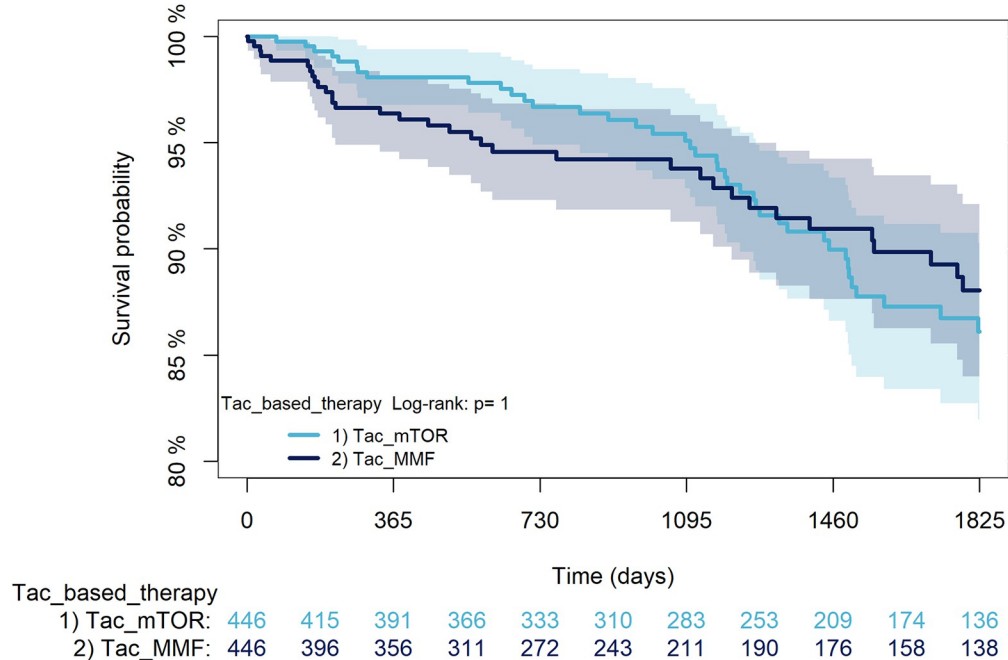

**Fig 13. Kaplan-Meier curves for survival according to mTORi or MMF within patients treated with TAC.**

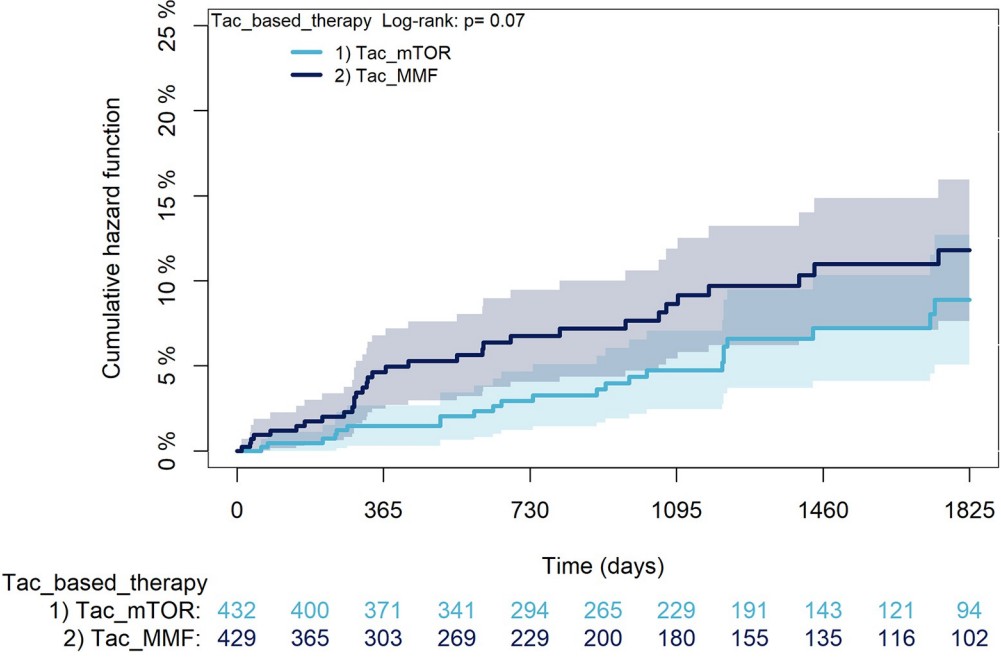

**Fig 14. Kaplan-Meier curves for rejection/graft loss according to mTORi or MMF within patients treated with TAC.**

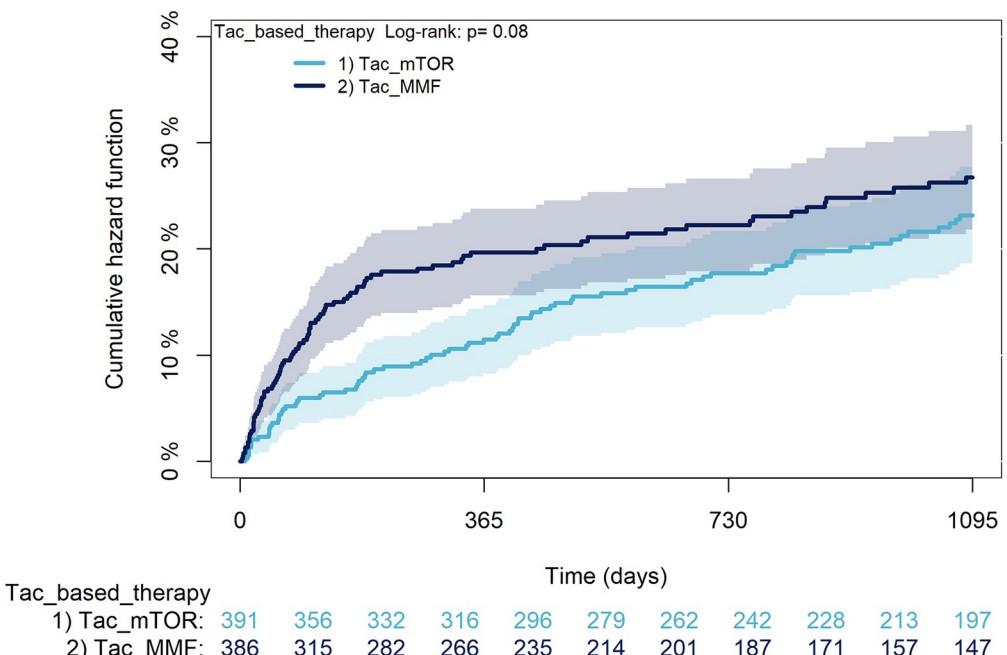

**Fig 15. Kaplan-Meier curves for severe infections according to mTORi or MMF within patients treated with TAC.**

subgroup. Furthermore, there was a tendency towards a higher risk of MACE for mTORi users aged 18–29 years and 30–59 years, with HRs almost reaching statistical significance, on the same time in the oldest subgroup HR for MACE was less than 1; this may indicate that particular attention should be paid in younger patients when using mTORi.

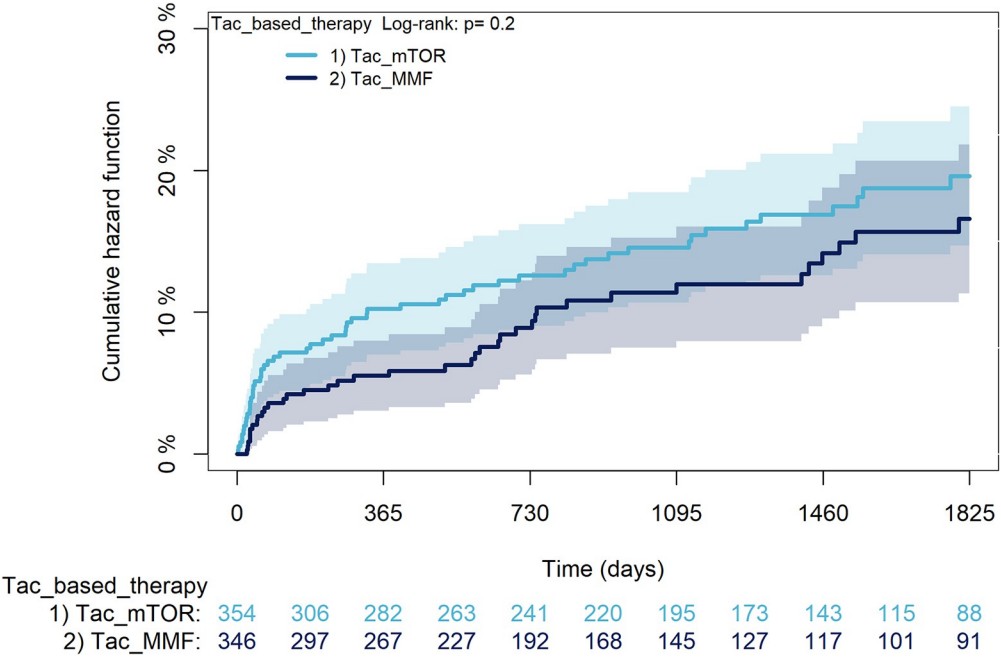

**Fig 16. Kaplan-Meier curves for diabetes according to mTORi or MMF within patients treated with TAC.**

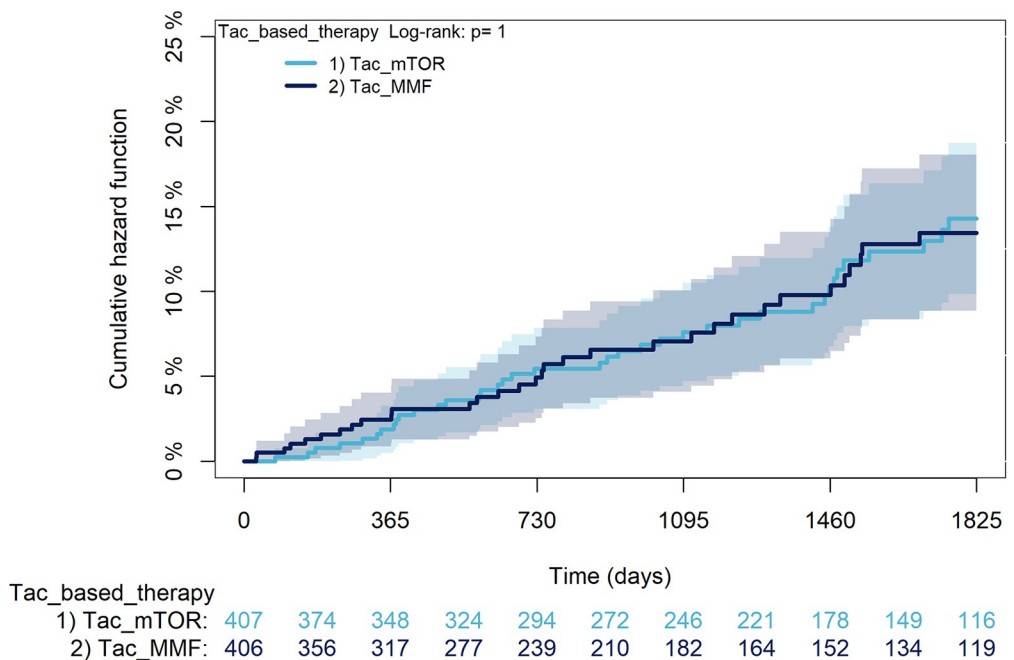

**Fig 17. Kaplan-Meier curves for cancer according to mTORi or MMF within patients treated with TAC.**

The proportions of patients assuming prednisone in combination with TAC and CsA were comparable (*TAC vs CsA*: 74.6% vs 73.3%); instead among mTORi-users the association with prednisone was higher than among MMF-users (*TAC+MMF vs TAC+mTORi*: 84.8% vs 65.9%), this did not translate into a different HR estimate when adjusting the model for prednisone-use.

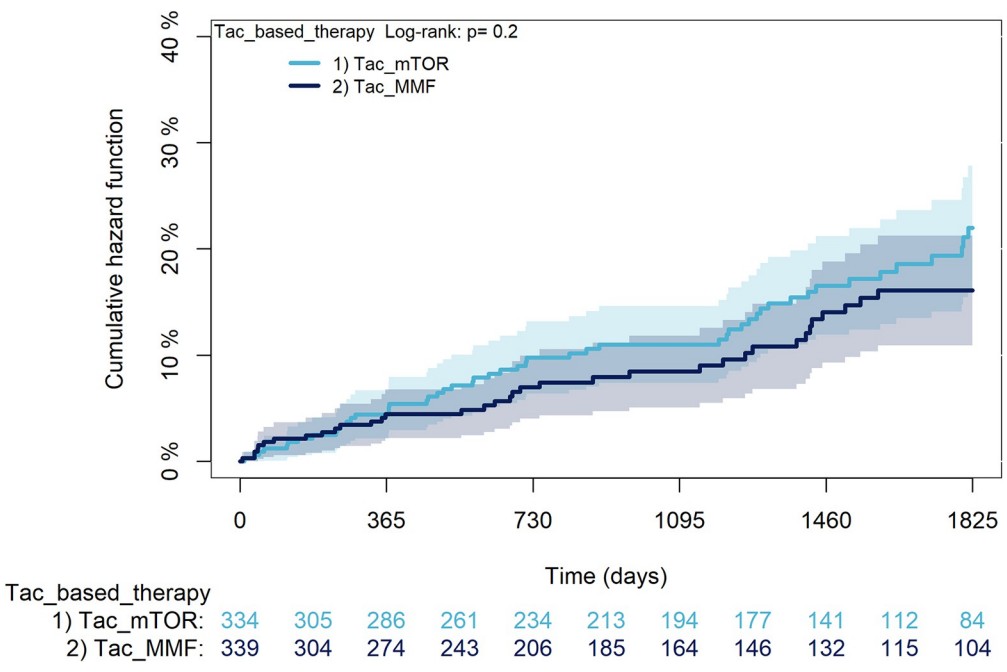

**Fig 18. Kaplan-Meier curves for MACA according to mTORi or MMF within patients treated with TAC.**

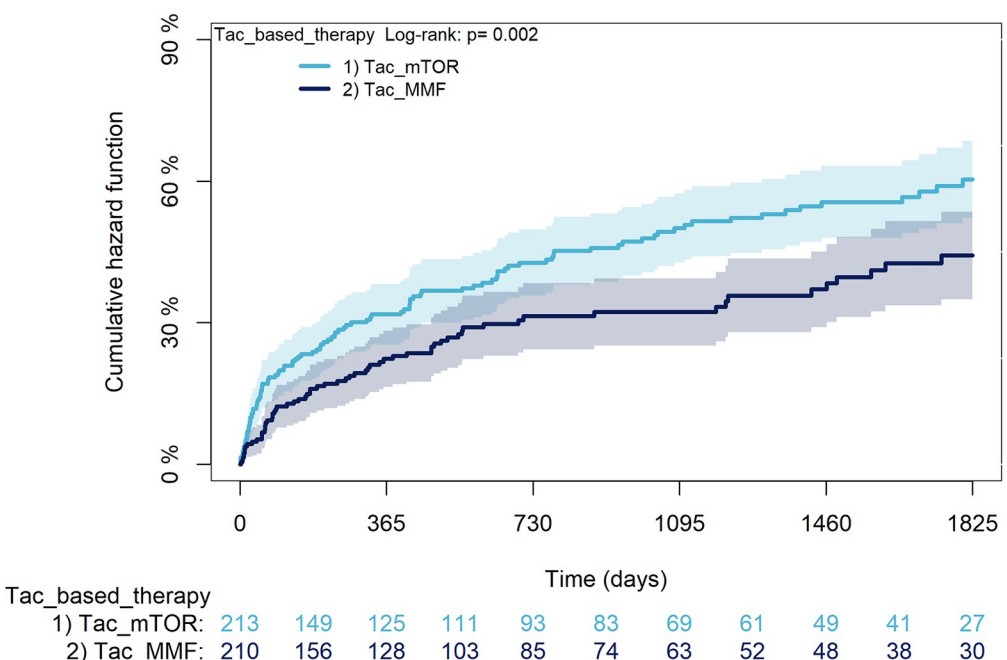

**Fig 19. Kaplan-Meier curves for statin use according to mTORi or MMF within patients treated with TAC. Note.** mTORi: Mammalian target of rapamycin inhibitors; MMF: Mycophenolate.

Further, the role that donor's previous infections and malignancies had in outcomes occurrence was explored: all combinations of therapies considered, were comparable in terms of frequency distribution of donor's infections (*TAC vs CsA*: 15.5% vs 16.2% *pvalue*: 0.715; *TAC +MMF vs TAC+mTORi*: 20.1% vs 15.7% *p-value*: 0.090) and cancer (*TAC vs CsA*: 2.4% vs 4.1% *pvalue*: 0.072; *TAC+MMF vs TAC+mTORi*: 5.7% vs 3.7% *p-value*: 0.159); also, no difference emerged in HR estimates after adjusting for both infections and cancer.

Finally, the adjustment of the model for DGF, did not result substantially in different estimated risks of rejection/graft loss (Table 2).

## Discussion

To the best of our knowledge, this is the largest multicentre observational study conducted in Europe comparing the effectiveness and safety profile of different immunosuppressive therapeutic regimens prescribed after kidney transplant.

The present work showed that in kidney recipients, TAC-based immunosuppressive therapy was significantly superior to CsA-based therapy in reducing rejection/graft loss and severe infections. At the same time, TAC was associated with a significantly higher risk of post transplantations diabetes mellitus compared to CsA. The combination of TAC with mTORi resulted in a higher risk of statin use compared to TAC and MMF.

The first result aligns with numerous recently published RCTs and meta-analyses evaluating TAC compared to CsA [25–29]. For instance, Ekberg and colleagues compared four immunosuppressive regimens (standard dose CsA, low dose CsA, low dose TAC and low dose Sirolimus) in 1,645 renal-transplant recipients; this study showed that the regimen including TAC was superior to all other treatment arms in terms of biopsy-proven acute rejection (BPAR) and allograft survival. The incidence of BPAR in the TAC group was approximately half compared to that in the low dose and standard dose CsA groups [25]. Another

**Table 2. Results of subgroup analysis and sensitivity analyses in the two comparison groups CsA vs TAC and mTORi vs MMF within patients treated with TAC.**

| | | Mortality | Rejection/Graft loss | Severe infections | Diabetes | Cancer | MACE | Use of statins |
|---|---|---|---|---|---|---|---|---|
| **HR (95%CI) CsA vs TAC** | **As treated (AT approach)** | 1.16 (0.71–1.91) | 1.47 (0.96–2.26) | 1.26 (0.99–1.61) | 0.65 (0.46–0.94) | 1.10 (0.74–1.64) | 1.36 (0.94–1.95) | 1.10 (0.86–1.42) |
| | **Age** | | | | | | | |
| | **18–29** | 1.04 (0.47–2.28) | 1.70 (1.05–2.76) | 1.17 (0.86–1.61) | 0.70 (0.43–1.13) | 0.95 (0.51–1.76) | 0.98 (0.61–1.60) | 1.17 (0.88–1.57) |
| | **30–59** | 1.01 (0.44–2.29) | 1.63 (0.99–2.66) | 1.11 (0.81–1.53) | 0.70 (0.43–1.14) | 0.92 (0.48–1.77) | 0.96 (0.59–1.55) | 1.13 (0.83–1.53) |
| | **60+** | 0.94 (0.56–1.58) | 1.38 (0.77–2.46) | 1.27 (0.88–1.84) | 0.60 (0.38–0.97) | 1.11 (0.69–1.77) | 1.27 (0.80–2.02) | 0.90 (0.62–1.32) |
| | **Prednisone use adjustment** | 1.05 (0.68–1.61) | 1.49 (1.04–2.13) | 1.28 (1.01–1.62) | 0.71 (0.51–1) | 1.14 (0.79–1.64) | 1.18 (0.85–1.64) | 1.08 (0.86–1.36) |
| | **Donor infections adjustment** | - | - | 1.27 (1.01–1.61) | - | - | - | - |
| | **Donor cancer adjustment** | - | - | - | - | 1.14 (0.79–1.64) | - | - |
| | **DGF adjustment** | - | 1.50 (1.04–2.14) | - | - | - | - | - |
| | | | | | | | | |
| **HR (95%CI) mTORi vs MMF** | **As treated (AT approach)** | 0.70 (0.39–1.26) | 0.24 (0.10–0.55) | 0.63 (0.44–0.90) | 1.45 (0.91–2.32) | 0.80 (0.43–1.50) | 0.82 (0.48–1.39) | 1.87 (1.33–2.64) |
| | **Age** | | | | | | | |
| | **18–29** | 0.81 (0.33–1.97) | 0.47 (0.23–0.99) | 0.67 (0.42–1.07) | 1.00 (0.58–1.73) | 0.71 (0.31–1.59) | 2.02 (0.95–4.29) | 1.64 (1.10–2.45) |
| | **30–59** | 0.83 (0.34–2.01) | 0.44 (0.20–0.93) | 0.69 (0.43–1.11) | 0.98 (0.56–1.73) | 0.73 (0.32–1.64) | 2.07 (0.97–4.40) | 1.68 (1.13–2.51) |
| | **60+** | 0.97 (0.57–1.67) | 0.87 (0.38–2.03) | 0.76 (0.51–1.14) | 1.80 (0.93–3.46) | 1.11 (0.62–2.01) | 0.94 (0.55–1.61) | 1.50 (0.92–2.45) |
| | **Prednisone use adjustment** | 1.01 (0.63–1.61) | 0.58 (0.33–1.00) | 0.74 (0.54–1.02) | 1.33 (0.87–2.03) | 1.02 (0.63–1.66) | 1.22 (0.78–1.90) | 1.78 (1.30–2.46) |
| | **Donor infections adjustment** | - | - | 0.77 (0.57–1.05) | - | - | - | - |
| | **Donor cancer adjustment** | - | - | - | - | 0.98 (0.61–1.57) | - | - |
| | **DGF adjustment** | - | 0.61 (0.36–1.05) | - | - | - | - | - |

**Note**. TAC: Tacrolimus; CsA: Cyclosporine; mTORi: Mammalian target of rapamycin inhibitors; MMF: Mycophenolate; HR: Hazard ratio; 95%CI: 95% confidence interval MACE: Major adverse cardiovascular events; DGF: Delayed Graft Function.

randomized open study, conducted in 50 transplant centres across seven European countries, showed that a composite endpoint consisting of graft loss, patient death and BPAR occurred more frequently in CsA patients than in TAC patients (42.8% with CsA and 25.9% with TAC; P<0.001) over a 2 years follow-up period [26]. Finally, three meta-analyses, including 30, 21 and 27 RCTs respectively, compared the efficacy of TAC with CsA as primary therapy after kidney transplantation, concluding that treating patients with TAC resulted in a substantial reduction in graft loss and acute rejection [7,8,30].

As far as we know, there is one recent study published in Brazil in 2020 that obtained results almost contrary to ours; the cohort study conducted by Gomes et colleagues, in fact, revealed better long-term outcomes (mortality rate, graft survival and re-transplantation) for CsA-based regimens versus TAC-based therapy [31].

Furthermore, in accordance with the evidence from literature [7,32–35] the higher risk of new-onset diabetes after transplantation (NODAT) associated with TAC-based regimens is

particularly relevant. This is especially noteworthy since the two groups were comparable in terms of corticosteroid use (TAC vs CsA: 74.6% vs 73.3%) and since they were matched by PS considering different risks factors associated with the onset of NODAT, such as BMI.

In our cohort, even though CsA users had higher risk of rejection/graft loss, they did not show an increased risk of mortality; this can be explained with the possibility for kidney recipients of returning in dialysis if a transplant fails.

On balance, when considering the use on CNIs, our work suggests that, as already emerged in published evidence [4,36], clinicians should prefer TAC as primary immunosuppressive therapy in kidney recipients because of its better risk-benefit profile. They may consider CsA as an alternative in patients with significant risk factors for diabetes. It also highlighted the fact that, during the study period, clinical practice in the four regions under study seemed to be in contrast with these previous findings, since CsA is still prescribed in a significant percentage of cases.

Regarding the second comparison, previous studies have already tried to establish benefits and risks associated with the use of mTORi in immunosuppressive regimens. Different studies investigated the use of mTORi in substitution of CNI [37–39] or in association with low doses of CNI as a kidney-sparing strategy [40,41]. Lower rates of acute rejection and renal disfunction have been demonstrated in these cases when compared to regimens with standard doses of CNI; however, when regimens including mTORi have been compared to the combination of MMF and CNI, results were contradictory [42,43].

In our cohort, the combination consisting in TAC and mTORi showed good results in terms of efficacy and safety when compared to the classical regimen based on TAC and MMF, with the exception of the use of statins, which was higher in the mTORi group. These results are consistent with those obtained in a recent randomized open-label two-arm study, the TRANSFORM trial [44], demonstrating that a regimen of everolimus with reduced TAC was non-inferior to MMF plus conventional CNI for a binary end point assessing immunosuppressive efficacy and preservation of graft function in kidney transplant patients at mild-to-moderate immunologic risk. Cucchiari and collegues in an observational study published in 2019 [45] confirmed and extended these findings, also considering high immunological risk recipients excluded from the trial. In those patients, results were even better in terms of rejection and graft function.

In terms of safety, the fact that everolimus has been associated with a decrease of CMV [46] infection represents a possible explanation for the lower rate of infections occurring in our cohort in the mTORi group compared to the MMF group (Fig 3B).

The observed risk of initiating statin therapy, along with existing body of evidence suggesting the role of mTORi in lipid homeostasis leading to hypercholesterolemia and hypertriglyceridemia [43,47], adds a noteworthy dimension to our findings. Despite this side-effect, mTORi appeared to contribute to the stabilization of the atherosclerotic plaque [48,49] and the reduction of left-ventricular hypertrophy [50]. This could potentially explain why the higher use of statins in our cohort did not translate into an increased risk of cardiovascular events. Furthermore, the KM estimate of the incident use of statins (Fig 5G) showed a statistically significant difference between the two curves from the beginning of the follow-up. This suggests that clinicians may have chosen to prescribe these medications as a preventive measure due to the well-known collateral effects of mTORi.

On the other hand, even if previous evidence highlighted the anti-neoplastic effects of mTORi [50–54], this aspect did not emerge from our analysis. This could be attributed to the relatively short follow-up duration (maximum of 5 years). It is also noteworthy that the use of mTORi has been associated with high discontinuation rate [55–57], representing a significant challenge in the real-world use of this class of drugs, which have often proven to be badly

**Table 3. Switching distribution among TAC and CsA users and mTORi and MMF users within patients treated with TAC.**

|  | TAC | CsA | TAC+MMF | TAC+mTOR |
|---|---|---|---|---|
| Switch N (%) | 41 (5.7%) | 67 (9.3%) | 44 (9.9%) | 163 (36.5%) |
| Time to first switch in months | 12 | 17 | 14 | 12 |
| Switch back N (%) | 4 (9.7%) | 5 (7.4%) | 13 (29.5%) | 33 (20.2%) |
| Time to switch back in months | 5 | 3 | 3 | 7 |

**Note**. CsA: Cyclosporine; TAC: Tacrolimus; mTORi: Mammalian target of rapamycin inhibitors; MMF: Mycophenolate.

tolerated. Various studies [58,59] demonstrated the association between mTORi treatment and impaired wound healing and cutaneous adverse events. These factors could become reasons for discontinuing the therapy, either due to the seriousness of some of these events or because of their social or functional impact. In our analysis, we only included major events that had required the use of health care services, it is possible that we did not account for minor side effects that may have contributed to therapy discontinuation and difficulties in observing some long-term outcomes in real life setting. In the cohort, the switching rate for mTORi group was higher than that for the MMF group (Table 3), and, interestingly, in 20.2% of cases we observed a switch back to the previous therapy. The consistency between ITT and AT analyses suggested that the switching rates did not change the risks of outcomes occurrence in the group considered. However, further studies should be conducted focusing on this aspect and taking into consideration minor collateral effects that may impact the medication management and patients' quality of life.

The main strength of this study lies in the availability of data on immunosuppressive dispensation from four regions, which are representative of Northern, Central and Southern Italy.

However, the study has some limitations. Firstly, being an observational study based on administrative data, we only considered drugs reimbursed by the healthcare system and there might be some imprecisions due to prescriptions from outside the region or privately purchased drugs. It is also possible an overestimation of drug use in case the drug is claimed at the pharmacy, but not actually taken by patients. However, the immunosuppressant medications considered in the analysis are rather expensive and the proportion of patients purchasing them privately can be considered negligible. Prednisone, which has a much lower cost and can be prescribed for a wide range of indications, may represent an exception to these considerations. To address this limitation, a sensitivity analysis was performed, and after adjusting for corticosteroids use, no different results emerged.

The administrative nature of the data also requires us to take into consideration the possibility of clinical unobserved factors influencing outcomes; although residual confounding may be present, the record linkage of data coming from the Italian national transplant centre (CNT) and the large cohort enrolled contribute to reinforcing the observed evidence. Nevertheless, further studies with access to more specific clinical data would enable the investigation of interesting aspects; for instance, it would be important to examine renal function at baseline and post-transplantation as an indicator of effectiveness that can assist clinicians in evaluating the prognosis.

Additionally, since the study relies on medication dispensation data where dosage is lacking, this information has not been taken into consideration. However, different minimization strategies can be applied to reduce the dosage of immunosuppressive maintenance drugs in

order to limit complications associated with them, especially CNIs [60]. Since the study involved a long monitoring period, it's plausible to assume that the different treatment groups encompassed patients assuming both low and high dosage therapies. This may have introduced a bias that may have impacted the incidence of certain outcomes (such as diabetes and infections); therefore, it would be worthwhile to conduct further studies to explore the issue of dosing and thus providing insights into the potential risks and benefits of different therapy combination at various dosages.

Further, we did not consider the use of Belatacept that was identified in NICE guidelines [4] as a possible option for maintenance therapy in kidney recipients; in fact, in Italy the use of this drug is limited to hospital settings and there is no information in our databases on medications used in hospitals.

Another limitation concerns induction therapy that was not assessed, due to the lack of this information in our databases. However, we expect that maintenance immunosuppression may play the major role in long-term outcomes considered in our study.

In conclusion, this study found that in a real-world setting, TAC-based immunosuppressive therapy has a significantly better effectiveness and safety profile when compared to CsA; particular attention should be paid to patients with medical history or risk factors for diabetes. The combination of TAC and mTORi may represent a valid alternative to the association of TAC and MMF, even if it is associated with an increased risk of incident use of statins. Further studies on this topic should be conducted to better define the role in therapy and prescribing recommendations of mTORi with respect to MMF. These results, on the one hand, may support policy makers and prescribers in clinical practice assisting them in choosing among different possible combinations as first-line therapy based on patients' characteristics; and, on the other hand, they also highlight the importance of better monitoring of different treatments to remodulate them based on emerging issues.

## Supporting information

**S1 Table. Codes ICDIX-CM for infections.**
(DOCX)

## Acknowledgments

*CESIT study group*: Alessandro C. Rosa, Marco Finocchietti, Francesca R Poggi, Maria Lucia Marino, Arianna Bellini, Claudia Marino, Ursula Kirchmayer, Nera Agabiti, Marina Davoli, Antonio Addis, Valeria Belleudi (Department of Epidemiology, Lazio Regional Health Service); Marco Massari, Stefania Spila Alegiani (Pharmacoepidemiology Unit, National Centre for Drug Research and Evaluation, Istituto Superiore di Sanità, Rome); Lucia Masiero, Andrea Ricci, Bedeschi Gaia, Francesca Puoti, Vito Sparacino, Pamela Fiaschetti, Silvia Trapani, Alessandra Oliveti, Daniela Peritore, Massimo Cardillo (Italian National Transplant Centre–Istituto Superiore di Sanità); Lorella Lombardozzi (Lazio Region); Silvia Pierobon, Eliana Ferroni, Maurizio Nordio, Manuel Zorzi (Veneto Region); Martina Zanforlini, Arianna Mazzone, Michele Ercolanoni, Giuseppe Piccolo, Andrea Angelo Nisic, Olivia Leoni (Lombardy Region); Stefano Ledda. Paolo Carta, Donatella Garau (Sardinia Region); Valentina Ientile, Luca L'Abbate (Messina University), Matilde Tanaglia, Gianluca Trifirò, Ugo Moretti (Verona University); Ersilia Lucenteforte (Pisa University).

## Author Contributions

**Conceptualization:** Valeria Belleudi.

**Data curation:** Alessandro Cesare Rosa, Marco Massari, Stefania Spila Alegiani, Lucia Masiero, Gaia Bedeschi, Silvia Pierobon, Stefano Ledda.

**Formal analysis:** Marco Finocchietti, Lucia Masiero.

**Funding acquisition:** Marina Davoli, Valeria Belleudi.

**Investigation:** Valeria Belleudi.

**Methodology:** Marco Finocchietti, Alessandro Cesare Rosa, Lucia Masiero, Ersilia Lucenteforte, Valeria Belleudi.

**Software:** Marco Massari, Stefania Spila Alegiani.

**Supervision:** Massimo Cardillo, Olivia Leoni, Donatella Garau, Marina Davoli, Antonio Addis, Valeria Belleudi.

**Writing – original draft:** Arianna Bellini, Marco Finocchietti.

**Writing – review & editing:** Marco Finocchietti, Alessandro Cesare Rosa, Maurizio Nordio, Eliana Ferroni, Marco Massari, Stefania Spila Alegiani, Lucia Masiero, Massimo Cardillo, Ersilia Lucenteforte, Giuseppe Piccolo, Olivia Leoni, Silvia Pierobon, Stefano Ledda, Donatella Garau, Marina Davoli, Antonio Addis, Valeria Belleudi.

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
