## [Decision Letter · Decision Letter 0]

11 Aug 2023

PONE-D-23-20920Effectiveness and safety of immunosuppressive regimens used as maintenance therapy in kidney transplantation: the CESIT study.PLOS ONE

Dear Dr. Belleudi,

Thank you for submitting your manuscript to PLOS ONE. After careful consideration, we feel that it has merit but does not fully meet PLOS ONE’s publication criteria as it currently stands. Therefore, we invite you to submit a revised version of the manuscript that addresses the points raised during the review process.

We look forward to receiving your revised manuscript.

Kind regards,

Yavuz - Ayar

Academic Editor

PLOS ONE

Journal Requirements:

Additional Editor Comments:

Dear Author/s

Greetings

Thank you for sharing your article with us. Major revision is requested as a result of the evaluations of experienced reviewers.

Best regards

Reviewers' comments:

Reviewer's Responses to Questions

**Comments to the Author**

1. Is the manuscript technically sound, and do the data support the conclusions?

Reviewer #1: Partly

Reviewer #2: Yes

2. Has the statistical analysis been performed appropriately and rigorously? 

Reviewer #1: I Don't Know

Reviewer #2: Yes

3. Have the authors made all data underlying the findings in their manuscript fully available?

Reviewer #1: Yes

Reviewer #2: Yes

4. Is the manuscript presented in an intelligible fashion and written in standard English?

Reviewer #1: Yes

Reviewer #2: Yes

5. Review Comments to the Author

Reviewer #1: The manuscript by Bellini and colleagues reports the results of an observational (registry) study on the use of CNI based maintenance therapies in kidney transplant recipients, using propensity score matching.

Their results confirm in a large cohort included over a 10 years period, important findings such as the superiority of tacrolimus as compared to CsA for graft outcomes and lower risk of diabetes as well as higher risk of dyslipidemia as defined by statin use for the combination tac+mTORi as compared to tac+MMF. The abstract mentions a potential protective effect of mTORi+Tac as compared to MMF+Tac that do not reach statistical significance and is not supported by RCTs (eg Transform).

Sensitivity analysis are performed accounting for meaningful clinical variables.

Major:

The nature of registry data significantly limits granularity of the analysis, being the most important outcome a combination of rejection/graft loss.

Can the author clarify the definition of this variable? Is rejection reliably recorded (Biopsy?discharge diagnosis? Receiving rescue treatment?) in this registry? In case not, only graft loss might be a more accurate outcome since it depends on clear events such as starting dialysis or retrasplantation. If, on the other hand, rejection IS reliably recorded, splitting the outcome in two might be interesting since graft loss is of course not related to alloimmune events in a large percentage of cases.

Minor:

Were recipients of other solid organ transplants excluded from the analysis?

It would be useful to report mean or median follow-up time of the cohort.

In line 119 page 5 ,HLA-DL DR?

Is the outcome “diabetes” restricted to post transplant? Please define in methods.

Does Table 2 show Hazard ratios? the note has maybe some incongruence with the table or are SMD shown here?

Reviewer #2: I thoroughly reviewed the paper and found it to be both interesting and pertinent. Despite the abundance of existing literature on the topic, it's crucial to acknowledge potential variations in data across different locations. Assessing the effectiveness and safety of various immunosuppressive regimens used in kidney transplantation adds significant value to the overall comprehension of the subject.

My comments and questions are as follows:

1) Page 2 line 39 - Abstract: The supposed benefit of mTORi usage, as it was not statistically significant, should not be stated in the abstract as a certainty. It could be described as non-inferiority instead.

2) Methods: Elaborate further on the specific parameters used to classify infections as severe, and provide a detailed description of the methodology employed to gather the necessary data for conducting the effectiveness analysis in relation to these severe infections.

3) Methods: why wasn't the type of donor considered in PS-matching?

4) Methods: Elaborate further on the specific parameters used to classify transplant hospital length as standard or prolonged

5) Page 5 line 119 - Correct spelling : length of transplant

6) Page 7 line 164 - Correct spelling : HLA- DR

7) Missing Explanation for Certain Adjustments: While the article adjusts for infections, malignancies, and delayed graft function (DGF), it might be helpful to provide a brief rationale for these adjustments and their relevance in the context of the study.

8) Graphics - enhance the resolution and select colors more effectively to improve visual clarity.

9) Results: Explore in the text information obtained from graphs 5A-F.

10) Page 15 line - Correct spelling : prednisone-use

11) The study fails to address a crucial aspect, namely medication dosages. Since the study relies on medication dispensation data, a substantial bias arises concerning the dosages of calcineurin inhibitors and mTORi. Considering the decade-long selection and monitoring period, it's reasonable to assume that patients in the tacrolimus group encompassed both low and high dosage users. This differentiation could significantly impact outcomes like the incidence of NODAT. Furthermore, distinguishing dosage variations in the tacrolimus mTORi/MPS -associated groups could provide more accurate insights into the actual extent of the potential benefits of mTORi usage. I recommend a deeper exploration of this bias within the article's limitations section.

While the study provides valuable insights into immunosuppressive regimens in kidney transplantation, it has some notable gaps. The lack of details about medication dosages, potential confounding factors, and inherent biases in retrospective data are concerning. Additionally, the interpretation of results and the discussion could have more critically addressed limitations and potential clinical implications of the findings. This could strengthen the validity and clinical applicability of the study's conclusions.

6. PLOS authors have the option to publish the peer review history of their article (what does this mean?). If published, this will include your full peer review and any attached files.

Reviewer #1: No

Reviewer #2: No

---

## [Author Response · Author response to Decision Letter 0]

19 Sep 2023

Reviewer #1: The manuscript by Bellini and colleagues reports the results of an observational (registry) study on the use of CNI based maintenance therapies in kidney transplant recipients, using propensity score matching.

Their results confirm in a large cohort included over a 10 years period, important findings such as the superiority of tacrolimus as compared to CsA for graft outcomes and lower risk of diabetes as well as higher risk of dyslipidemia as defined by statin use for the combination tac+mTORi as compared to tac+MMF. The abstract mentions a potential protective effect of mTORi+Tac as compared to MMF+Tac that do not reach statistical significance and is not supported by RCTs (eg Transform).

Sensitivity analysis are performed accounting for meaningful clinical variables.

Major:

The nature of registry data significantly limits granularity of the analysis, being the most important outcome a combination of rejection/graft loss.

Can the author clarify the definition of this variable? Is rejection reliably recorded (Biopsy?discharge diagnosis? Receiving rescue treatment?) in this registry? In case not, only graft loss might be a more accurate outcome since it depends on clear events such as starting dialysis or retrasplantation. If, on the other hand, rejection IS reliably recorded, splitting the outcome in two might be interesting since graft loss is of course not related to alloimmune events in a large percentage of cases.

We thank the reviewer for the comment. Information on transplant rejection was detected by the Transplant Information System an infrastructure for the management of data related to the activity of the National Transplant Network, established and regulated by Italian Laws (n. 91 of April 1, 1999). The SIT collecting data on donation, allocation and transplantation of organ including transplanted organ quality, through SIT it is possible to ensure transparency and traceability of the donation, retrieval and transplantation processes. Specifically, data on organ rejection is detected directly by clinicians following a finding of impairment of transplanted organ function due to histologically documented immunological causes. Furthermore, data quality qudits and checks are frequently performed. We added these details in the method section.

Minor:

Were recipients of other solid organ transplants excluded from the analysis?

Yes, we excluded multi-organ transplant recipients. We added details on this point in the method section and in the flow-chart (Figure 1). More details about the selection of the cohort can be found in the previous paper:

Belleudi V, Rosa AC, Finocchietti M, Poggi FR, Marino ML, Massari M,

Spila Alegiani S, Masiero L, Ricci A, Bedeschi G, Puoti F, Cardillo M, Pierobon S, Nordio M, Ferroni E, Zanforlini M, Piccolo G, Leone O, Ledda S, Carta P, Garau D, Lucenteforte E, Davoli M and Addis A, CESIT Study Group (2022), An Italian multicentre distributed data research network to study the use, effectiveness, and safety of immunosuppressive drugs in transplant patients: Framework and perspectives of the CESIT project. Front. Pharmacol. 13:959267.

It would be useful to report mean or median follow-up time of the cohort. 

We appreciate the reviewer's suggestion, we added this information in the result section.

In line 119 page 5 ,HLA-DL DR? 

You are right we have corrected the typo

Is the outcome “diabetes” restricted to post transplant? Please define in methods.

As reported in the methods section “for each outcome, only patients who were at risk of developing the outcome for the first time were considered” (Page 6). To clarify this point, we modified the manuscript by referring to new-onset diabetes.

Does Table 2 show Hazard ratios? the note has maybe some incongruence with the table or are SMD shown here?

Yes, Table 2 shows HR. To clarify this point, we added description in the table. 

Reviewer #2: I thoroughly reviewed the paper and found it to be both interesting and pertinent. Despite the abundance of existing literature on the topic, it's crucial to acknowledge potential variations in data across different locations. Assessing the effectiveness and safety of various immunosuppressive regimens used in kidney transplantation adds significant value to the overall comprehension of the subject.

My comments and questions are as follows:

1) Page 2 line 39 - Abstract: The supposed benefit of mTORi usage, as it was not statistically significant, should not be stated in the abstract as a certainty. It could be described as non-inferiority instead. 

We agree with you. We modified the abstract following your suggestion.

2) Methods: Elaborate further on the specific parameters used to classify infections as severe, and provide a detailed description of the methodology employed to gather the necessary data for conducting the effectiveness analysis in relation to these severe infections.

The infection-related outcome was based on only severe infections, which were defined as those resulting in hospitalization. We have included in S1 Table the ICD-IXCM codes selected for this outcome and in References section some papers on the basis of which we made the choice of codes.

3) Methods: why wasn't the type of donor considered in PS-matching?

We apologize to you, we forgot to indicate this in the list of variables considered, although we had included it in PS-matching (as shown in Figure 2 A/B). We corrected the method section.

4) Methods: Elaborate further on the specific parameters used to classify transplant hospital length as standard or prolonged

Prolonged hospitalization was defined as a length of stay equal to or greater than the 75th percentile of length of stay of all participants

5) Page 5 line 119 - Correct spelling : length of transplant 

You are right we have corrected the typo

6) Page 7 line 164 - Correct spelling : HLA- DR 

You are right we have corrected the typo

7) Missing Explanation for Certain Adjustments: While the article adjusts for infections, malignancies, and delayed graft function (DGF), it might be helpful to provide a brief rationale for these adjustments and their relevance in the context of the study.

Thank you for your suggestion, we have modified the text and the bibliography by inserting a brief rationale regarding adjustments and by adding some previous work on which we had based our choice of adjustments

8) Graphics - enhance the resolution and select colors more effectively to improve visual clarity.

We modified resolution and colors to improve the graphics. We are available to consider further changes with the editor if the paper will be accepted for publication.

9) Results: Explore in the text information obtained from graphs 5A-F.

We added a commentary paragraph on graphs in the result section.

10) Page 15 line - Correct spelling: prednisone-use

You are right we have corrected the typo

11) The study fails to address a crucial aspect, namely medication dosages. Since the study relies on medication dispensation data, a substantial bias arises concerning the dosages of calcineurin inhibitors and mTORi. Considering the decade-long selection and monitoring period, it's reasonable to assume that patients in the tacrolimus group encompassed both low and high dosage users. This differentiation could significantly impact outcomes like the incidence of NODAT. Furthermore, distinguishing dosage variations in the tacrolimus mTORi/MPS -associated groups could provide more accurate insights into the actual extent of the potential benefits of mTORi usage. I recommend a deeper exploration of this bias within the article's limitations section. While the study provides valuable insights into immunosuppressive regimens in kidney transplantation, it has some notable gaps. The lack of details about medication dosages, potential confounding factors, and inherent biases in retrospective data are concerning. Additionally, the interpretation of results and the discussion could have more critically addressed limitations and potential clinical implications of the findings. This could strengthen the validity and clinical applicability of the study's conclusions.

Thank you for your comment, this is indeed an important point of the limitations of the study and an important element to be explored with future investigations, we have included a sentence in the limitations section to emphasise the lack of consideration of dose and the possible bias concerning this.

---

## [Decision Letter · Decision Letter 1]

12 Oct 2023

PONE-D-23-20920R1Effectiveness and safety of immunosuppressive regimens used as maintenance therapy in kidney transplantation: the CESIT study.PLOS ONE

Dear Dr. Belleudi,

Thank you for submitting your manuscript to PLOS ONE. After careful consideration, we feel that it has merit but does not fully meet PLOS ONE’s publication criteria as it currently stands. Therefore, we invite you to submit a revised version of the manuscript that addresses the points raised during the review process.

We look forward to receiving your revised manuscript.

Kind regards,

Yavuz - Ayar

Academic Editor

PLOS ONE

Additional Editor Comments:

Dear Author/s

Greetings

After the evaluation made by reviewers, a major revision decision was made for the article.

Best regards

Reviewers' comments:

Reviewer's Responses to Questions

**Comments to the Author**

1. If the authors have adequately addressed your comments raised in a previous round of review and you feel that this manuscript is now acceptable for publication, you may indicate that here to bypass the “Comments to the Author” section, enter your conflict of interest statement in the “Confidential to Editor” section, and submit your "Accept" recommendation.

Reviewer #3: All comments have been addressed

Reviewer #4: All comments have been addressed

Reviewer #5: (No Response)

Reviewer #6: (No Response)

Reviewer #7: (No Response)

Reviewer #8: (No Response)

2. Is the manuscript technically sound, and do the data support the conclusions?

Reviewer #3: Partly

Reviewer #4: Yes

Reviewer #5: Yes

Reviewer #6: Yes

Reviewer #7: Yes

Reviewer #8: (No Response)

3. Has the statistical analysis been performed appropriately and rigorously? 

Reviewer #3: Yes

Reviewer #4: I Don't Know

Reviewer #5: Yes

Reviewer #6: Yes

Reviewer #7: Yes

Reviewer #8: (No Response)

4. Have the authors made all data underlying the findings in their manuscript fully available?

Reviewer #3: Yes

Reviewer #4: Yes

Reviewer #5: Yes

Reviewer #6: Yes

Reviewer #7: Yes

Reviewer #8: (No Response)

5. Is the manuscript presented in an intelligible fashion and written in standard English?

Reviewer #3: Yes

Reviewer #4: Yes

Reviewer #5: No

Reviewer #6: Yes

Reviewer #7: Yes

Reviewer #8: (No Response)

6. Review Comments to the Author

Reviewer #3: The manuscript evaluated the effectiveness and safety by comparison between cyclosporine and tacrolimus-based therapies, Tacrolimus + mTORi and Tacrolimus + MMF in kidney transplant recipients and reported the results of real-world data on the use of CNI based maintenance therapies in kidney transplant recipients.

1.It would be very important to report baseline renal function ( serum creatinine ,GFR or eGFR ) and urine protein of transplant kidney.

2.The outcome considered in the manuscript were mortality and transplant reject/graft failure for effectiveness analysis, and incidence of severe infections, cancer, diabetes and statin use for safety analysis. The outcome should consider renal function progression. ≥50% decline in eGFR should be added to the primary outcome for effectiveness analysis.

Reviewer #4: Dear authors,

I think the revised form of the manuscript is quite improved. I will hava e few additional comments.

The use of mTOR inhibitors as part of initial maintenance therapy is usually limited by early posttransplant complications (delayed allograft function, poor wound healing, and an increased incidence of lymphoceles) associated with these agents. Did you have any data about this? Were mTOR inhibitors used in these patients as a part of initial maintenance therapy?

Were the ones under the treatment of azathioprine excluded?

Lack of data about induction treatment is an important limitation. Data whether target levels of CNI’s were reached or not was also lacking.

Did you know BK virus infection prevalance in these different groups?

P64 line 99 LAR should be explained. (legally acceptable representative)

Table 1 “sovrappeso” should be corrected.

Best regards

Reviewer #5: This is an interesting study that tried to compare the outcomes of kidney transplant patients whether they were on tacrolimus or cyclosporine and whether they were on MMF or mTORi. They used the PS matching to make these comparisons on a real-world cohort.

Although the PS matching is the best way to reduce any confounding bias, the authors lost almost half of the sample because they could not match them. When we look at Fig 2A we can see that the differences between those matched or not for the comparison of TACROLIMUS to CYCLOSPORINE do not include cancer, comorbidities, metabolic parameters, etc..however, in Fig2B, the differences are more important especially when it comes to cancer that is one of the indicatiosn to switch to mTORi. Therefore, authors need to align their conclusions with this important limitation of the PS matching selection.

My other comment is that this manuscript should be revised by a native English speaker (some sentences are really hard to read).

Reviewer #6: Really interesting stuff.

Keywords: Please uniform. Either all in lowercase, or all in uppercase. Choose and be consistent.

Is 'raw data' available?

This is very important, the verifiability and repeatability of the research. It is unclear whether or not.

I agree that the outcome of rejection must be histologically documented.

P2 L38: Please, write 'risk of rejection/' instead of 'reject/'.

P4 L77: Avoid abbreviations at the very beginning of the sentence.

etc.

Go through the text a few more times, there are still some lexical and grammatical errors.

P4 L81: There is still the question of correction of IS therapy in the event of malignancy, conversion to mTOR, etc., where even some of our national transplant experts cannot give an unequivocal answer or conclusion as to what to do.

As you said - guidelines are one thing, life is another.

Reviewer #7: In the current manuscript, Bellini et al. reveal the effectiveness and safety profile of different immunosuppressive regimens in kidney transplantation. A large amount of multicenter data was collected and statistical analysis was performed. Their results indicated that tacrolimus-based immunosuppressive therapy appeared to be superior to cyclosporine in reducing rejection and severe infections. Besides, the combination of tacrolimus and mTORi may represent a valid alternative to the association with mycophenolate. These results are very important to improve graft survival and reduce acute rejection.

1.Mortality and transplant reject/graft failure were defined as outcomes of this study. Is it also possible to include renal function like eGFR as the outcomes? To use renal function as an indicator of effectiveness could help clinicians to evaluate the prognosis. I recommend a deeper exploration of this outcome if it is possible.

2.Please unify the wording, such as tumor and cancer.

3.The study relies on medication dispensation data where dosage is lacking. However, medication dosage is a crucial factor to assess the adverse outcome. I noticed hospital information system and co-payment exemption registry. Is it possible to acquire the dosage information by fee items or claims?

4.Method - Given the new onset of diabetes and statin use are ones of the outcome in the study, it’s inappropriate to have the history of diabetes and statin use at baseline as covariates. Please excluded patients with these two factors from the beginning.

5.Result - Table 1 tells several indicators’ SMD value of comparison pairs between TAC+MMF and TAC+mTORi ≥0.1. The indicators should be included as confounders in Cox models to assess the independent effect of exposures.

6.Result - The 112 (20%) patients of TAC+mTORi group were excluded according to the data after propensity score matching, which may result in sampling bias. Please use one to more matching or provide the outcome of the excluded data.

7.Graphics - Figure 2A and 2B were seen unclear here. Please provide clearer figures.

Reviewer #8: (No Response)

7. PLOS authors have the option to publish the peer review history of their article (what does this mean?). If published, this will include your full peer review and any attached files.

Reviewer #3: No

Reviewer #4: No

Reviewer #5: No

Reviewer #6: No

Reviewer #7: No

Reviewer #8: No

---

## [Author Response · Author response to Decision Letter 1]

26 Oct 2023

Reviewer #3: The manuscript evaluated the effectiveness and safety by comparison between cyclosporine and tacrolimus-based therapies, Tacrolimus + mTORi and Tacrolimus + MMF in kidney transplant recipients and reported the results of real-world data on the use of CNI based maintenance therapies in kidney transplant recipients.

1.It would be very important to report baseline renal function ( serum creatinine ,GFR or eGFR ) and urine protein of transplant kidney.

Thank you for the suggestion since the work in based on administrative data we had limited access to clinical information about our cohort. Data regarding baseline renal function and proteinuria were not available within the SIT; we had some data on renal function after transplantation (GFR). However, we chose not to add renal function as an effectiveness outcome because the variable resulted in a high rate of missing values (exceeding 80% for some of the years considered). We are aware that this is an important limit of the study and we have rephrased the discussion, emphasizing the importance of investigating this aspect in future studies.

2.The outcome considered in the manuscript were mortality and transplant reject/graft failure for effectiveness analysis, and incidence of severe infections, cancer, diabetes and statin use for safety analysis. The outcome should consider renal function progression. ≥50% decline in eGFR should be added to the primary outcome for effectiveness analysis.

As previously mentioned, the clinical information about renal function is not available. The study is observational, based on administrative claims and CNT data, this limitation is addressed in the discussion.

Reviewer #4: Dear authors,

I think the revised form of the manuscript is quite improved. I will hava e few additional comments.

The use of mTOR inhibitors as part of initial maintenance therapy is usually limited by early posttransplant complications (delayed allograft function, poor wound healing, and an increased incidence of lymphoceles) associated with these agents. Did you have any data about this? Were mTOR inhibitors used in these patients as a part of initial maintenance therapy?

No, we don’t have this data because, through administrative data, we can only track side effects that require the use of healthcare services; we have outlined this limitation in the discussion section. However, our study revealed that in some medical centers, mTORi are prescribed in 60% of cases, and it is unlikely that all of this usage can be attributed to these complications.

Were the ones under the treatment of azathioprine excluded?

Yes, we excluded these patients. We added this information in the method section.

Lack of data about induction treatment is an important limitation. Data whether target levels of CNI’s were reached or not was also lacking.

No, the nature of administrative data precludes the acquisition of detailed clinical patient information. Nevertheless, we believe that this work can enhance the understanding of maintenance immunosuppression and its use. As reported in the discussion section, the result observed in our unselected population are similar to those seen in trials. Despite these limitations in data availability, we do not believe that these factors have introduced distortions in our study.

Did you know BK virus infection prevalance in these different groups? 

The ICD-9CM come for tracking BK virus infection is 079.89, which corresponds to “Other specified viral infection”. Since it is not an exclusive and specific code, we have decided not to use it for tracking infections. However, this aspect is certainly interesting and we will consider it for future project developments..

P64 line 99 LAR should be explained. (legally acceptable representative)

Thank you, we have explained the acronym.

Table 1 “sovrappeso” should be corrected. 

We have translated the word.

Reviewer #5: This is an interesting study that tried to compare the outcomes of kidney transplant patients whether they were on tacrolimus or cyclosporine and whether they were on MMF or mTORi. They used the PS matching to make these comparisons on a real-world cohort.

Although the PS matching is the best way to reduce any confounding bias, the authors lost almost half of the sample because they could not match them. When we look at Fig 2A we can see that the differences between those matched or not for the comparison of TACROLIMUS to CYCLOSPORINE do not include cancer, comorbidities, metabolic parameters, etc..however, in Fig2B, the differences are more important especially when it comes to cancer that is one of the indicatiosn to switch to mTORi. Therefore, authors need to align their conclusions with this important limitation of the PS matching selection.

My other comment is that this manuscript should be revised by a native English speaker (some sentences are really hard to read).

We lost half of the sample, but this reduction was mainly due to the fact that there were only 787 CsA users; hence, the patients for whom no match was found were only 64 (787-723). The study aimed to analyze the association between therapies and outcomes and we chose to use statistical methods that allowed for internally valid results, even though this may result in a reduced external validity.

We added a reference in the text where Rubin DB and colleagues demonstrated that propensity score matching is the best way to eliminate differences between groups (42).

• Rosenbaum PR, Rubin DB. The central role of the propensity score in observational studies for causal effects. Biometrika. 1983;70:41–55.

• Rubin DB. Using multivariate matched sampling and regression adjustment to control bias in observational studies. J Am Stat Assoc. 1979;74:318–328. [Google Scholar]

• Rubin DB, Thomas N. Matching using estimated propensity scores: relating theory to practice. Biometrics. 1996;52:249–264. [PubMed] [Google Scholar]

Figures 2a and 2b demonstrate that after matching all the characteristics between the groups are balanced.

Reviewer #6: Really interesting stuff.

Keywords: Please uniform. Either all in lowercase, or all in uppercase. Choose and be consistent.

Is 'raw data' available?

This is very important, the verifiability and repeatability of the research. It is unclear whether or not. 

As reported in the manuscript “The data that support the findings of this study are available from the Italian regions participating to CESIT study but restrictions apply to the availability of these data, which were used under license (as by third-party sources) for the current study, and so are not publicly available. However, data are available with permission of Italian regions, which are the data owner. The non-author contact information to which data requests may be sent is: project.cesit@gmail.com.” 

Specific request will be evaluated.

I agree that the outcome of rejection must be histologically documented.

Ok. We reported this information in the method section.

P2 L38: Please, write 'risk of rejection/' instead of 'reject/'.

Ok. 

P4 L77: Avoid abbreviations at the very beginning of the sentence.

etc.

Ok. 

Go through the text a few more times, there are still some lexical and grammatical errors.

Ok. 

We have corrected the errors.

P4 L81: There is still the question of correction of IS therapy in the event of malignancy, conversion to mTOR, etc., where even some of our national transplant experts cannot give an unequivocal answer or conclusion as to what to do.

As you said - guidelines are one thing, life is another.

This is an interesting question, but the answer of this concern was out of scope of our work.

Reviewer #7: In the current manuscript, Bellini et al. reveal the effectiveness and safety profile of different immunosuppressive regimens in kidney transplantation. A large amount of multicenter data was collected and statistical analysis was performed. Their results indicated that tacrolimus-based immunosuppressive therapy appeared to be superior to cyclosporine in reducing rejection and severe infections. Besides, the combination of tacrolimus and mTORi may represent a valid alternative to the association with mycophenolate. These results are very important to improve graft survival and reduce acute rejection.

1.Mortality and transplant reject/graft failure were defined as outcomes of this study. Is it also possible to include renal function like eGFR as the outcomes? To use renal function as an indicator of effectiveness could help clinicians to evaluate the prognosis. I recommend a deeper exploration of this outcome if it is possible.

Thank you for your suggestion. Even though we agree that renal function would be interesting to investigate in our cohort, we have not included it as one of the outcomes because the study is based on administrative data and we had limited access to information on renal function. The variable related to GFR resulted in a high rate of missing values (exceeding 80% for some of the years considered). We emphasized this aspect in the discussion section.

2. Please unify the wording, such as tumor and cancer.

Thank you for the suggestion, we have replaced the word tumor with cancer in the text.

3.The study relies on medication dispensation data where dosage is lacking. However, medication dosage is a crucial factor to assess the adverse outcome. I noticed hospital information system and co-payment exemption registry. Is it possible to acquire the dosage information by fee items or claims?

Dosage in administrative data can be obtained using “Defined Daily Dose” (DDD), which is the assumed average maintenance dose per day for a drug used for its main indication in adults. However, it is likely that in the context of transplant the therapeutic doses used in clinical practice will be different from the DDD and will depend on individual patient characteristics. This aspect was already mentioned in the limits, we have further clarified it in the discussion.

4. Method - Given the new onset of diabetes and statin use are ones of the outcome in the study, it’s inappropriate to have the history of diabetes and statin use at baseline as covariates. Please excluded patients with these two factors from the beginning.

We had already excluded patients with history of diabetes and statin use from the analysis; we have rephrased the sentence in the methods section for better clarity (L140-142)

5. Result - Table 1 tells several indicators’ SMD value of comparison pairs between TAC+MMF and TAC+mTORi ≥0.1. The indicators should be included as confounders in Cox models to assess the independent effect of exposures.

Ok. As reported in the method we included in the model adjustment for propensity score value 

6. Result - The 112 (20%) patients of TAC+mTORi group were excluded according to the data after propensity score matching, which may result in sampling bias. Please use one to more matching or provide the outcome of the excluded data.

The study’s objective was to examine the relationship between therapies and the outcomes consideres; we opted for statistical methods that prioritize internal validity, albeit potentially reducing external validity.

Rubin DB and colleagues demonstrated that propensity score matching is the best way to eliminate differences between groups (42).

• Rosenbaum PR, Rubin DB. The central role of the propensity score in observational studies for causal effects. Biometrika. 1983;70:41–55.

• Rubin DB. Using multivariate matched sampling and regression adjustment to control bias in observational studies. J Am Stat Assoc. 1979;74:318–328. [Google Scholar]

• Rubin DB, Thomas N. Matching using estimated propensity scores: relating theory to practice. Biometrics. 1996;52:249–264. [PubMed] [Google Scholar]

We added this paper in the references.

7. Graphics - Figure 2A and 2B were seen unclear here. Please provide clearer figures.

OK

Reviewer #8: (No Response)

---

## [Editor Report · Decision Letter 2]

31 Oct 2023

PONE-D-23-20920R2Effectiveness and safety of immunosuppressive regimens used as maintenance therapy in kidney transplantation: the CESIT study.PLOS ONE

Dear Dr. Belleudi,

Thank you for submitting your manuscript to PLOS ONE. After careful consideration, we feel that it has merit but does not fully meet PLOS ONE’s publication criteria as it currently stands. Therefore, we invite you to submit a revised version of the manuscript that addresses the points raised during the review process.

We look forward to receiving your revised manuscript.

Kind regards,

Yavuz - Ayar

Academic Editor

PLOS ONE

Additional Editor Comments:

Dear Author/s

Greetings

After the evaluations, a major revision decision has been made for your article.

Best regards

---

## [Author Response · Author response to Decision Letter 2]

8 Nov 2023

Reviewer #3: The manuscript evaluated the effectiveness and safety by comparison between cyclosporine and tacrolimus-based therapies, Tacrolimus + mTORi and Tacrolimus + MMF in kidney transplant recipients and reported the results of real-world data on the use of CNI based maintenance therapies in kidney transplant recipients.

1.It would be very important to report baseline renal function ( serum creatinine ,GFR or eGFR ) and urine protein of transplant kidney.

Thank you for the suggestion since the work in based on administrative data we had limited access to clinical information about our cohort. Data regarding baseline renal function were not available within the SIT; we had some data on renal function after transplantation (GFR). However, we chose not to add renal function as an effectiveness outcome because the variable resulted in a high rate of missing values (exceeding 80% for some of the years considered). We are aware that this is an important limit of the study and we have rephrased the discussion, emphasizing the importance of investigating this aspect in future studies.

2.The outcome considered in the manuscript were mortality and transplant reject/graft failure for effectiveness analysis, and incidence of severe infections, cancer, diabetes and statin use for safety analysis. The outcome should consider renal function progression. ≥50% decline in eGFR should be added to the primary outcome for effectiveness analysis.

As previously mentioned, the clinical information about renal function is not available. The study is observational, based on administrative claims and CNT data, this limitation is addressed in the discussion.

Reviewer #4: Dear authors,

I think the revised form of the manuscript is quite improved. I will hava e few additional comments.

The use of mTOR inhibitors as part of initial maintenance therapy is usually limited by early posttransplant complications (delayed allograft function, poor wound healing, and an increased incidence of lymphoceles) associated with these agents. Did you have any data about this? Were mTOR inhibitors used in these patients as a part of initial maintenance therapy?

No, we don’t have this data. We have outlined this limitation in the discussion section. However, our study revealed that in some medical centers, mTORi are prescribed in 60% of cases, and it is unlikely that all of this usage can be attributed to these complications.

Were the ones under the treatment of azathioprine excluded?

Yes, we excluded these patients. We added this information in the method section.

Lack of data about induction treatment is an important limitation. Data whether target levels of CNI’s were reached or not was also lacking.

No, the nature of administrative data precludes the acquisition of detailed clinical patient information. Nevertheless, we believe that this work can enhance the understanding of maintenance immunosuppression and its use. As reported in the discussion section, the result observed in our unselected population are similar to those seen in trials. Despite these limitations in data availability, we do not believe that these factors have introduced distortions in our study.

Did you know BK virus infection prevalance in these different groups? 

The ICD-9CM come for tracking BK virus infection is 079.89, which corresponds to “Other specified viral infection”. Since it is not an exclusive and specific code, we have decided not to use it for tracking infections. However, this aspect is certainly interesting and we will consider it for future project developments.

P64 line 99 LAR should be explained. (legally acceptable representative)

Thank you, we have explained the acronym.

Table 1 “sovrappeso” should be corrected. 

We have translated the word.

Reviewer #5: This is an interesting study that tried to compare the outcomes of kidney transplant patients whether they were on tacrolimus or cyclosporine and whether they were on MMF or mTORi. They used the PS matching to make these comparisons on a real-world cohort.

Although the PS matching is the best way to reduce any confounding bias, the authors lost almost half of the sample because they could not match them. When we look at Fig 2A we can see that the differences between those matched or not for the comparison of TACROLIMUS to CYCLOSPORINE do not include cancer, comorbidities, metabolic parameters, etc..however, in Fig2B, the differences are more important especially when it comes to cancer that is one of the indicatiosn to switch to mTORi. Therefore, authors need to align their conclusions with this important limitation of the PS matching selection.

My other comment is that this manuscript should be revised by a native English speaker (some sentences are really hard to read).

We lost half of the sample, but this reduction was mainly due to the fact that there were only 787 CsA users; hence, the patients for whom no match was found were only 64 (787-723). The study aimed to analyze the association between therapies and outcomes and we chose to use statistical methods that allowed for internally valid results, even though this may result in a reduced external validity.

We added a reference in the text where Rubin DB and colleagues demonstrated that propensity score matching is the best way to eliminate differences between groups (42).

• Rosenbaum PR, Rubin DB. The central role of the propensity score in observational studies for causal effects. Biometrika. 1983;70:41–55.

• Rubin DB. Using multivariate matched sampling and regression adjustment to control bias in observational studies. J Am Stat Assoc. 1979;74:318–328. [Google Scholar]

• Rubin DB, Thomas N. Matching using estimated propensity scores: relating theory to practice. Biometrics. 1996;52:249–264. [PubMed] [Google Scholar]

Figures 2a and 2b demonstrate that after matching all the characteristics between the groups are balanced.

Reviewer #6: Really interesting stuff.

Keywords: Please uniform. Either all in lowercase, or all in uppercase. Choose and be consistent.

Is 'raw data' available?

This is very important, the verifiability and repeatability of the research. It is unclear whether or not. 

As reported in the manuscript “The data that support the findings of this study are available from the Italian regions participating to CESIT study but restrictions apply to the availability of these data, which were used under license (as by third-party sources) for the current study, and so are not publicly available. However, data are available with permission of Italian regions, which are the data owner. The non-author contact information to which data requests may be sent is: project.cesit@gmail.com.” 

Specific request will be evaluated.

I agree that the outcome of rejection must be histologically documented.

Ok. We reported this information in the method section.

P2 L38: Please, write 'risk of rejection/' instead of 'reject/'.

Ok. 

P4 L77: Avoid abbreviations at the very beginning of the sentence.

etc.

Ok. 

Go through the text a few more times, there are still some lexical and grammatical errors.

Ok. 

We have corrected the errors.

P4 L81: There is still the question of correction of IS therapy in the event of malignancy, conversion to mTOR, etc., where even some of our national transplant experts cannot give an unequivocal answer or conclusion as to what to do.

As you said - guidelines are one thing, life is another.

This is an interesting question, but the answer of this concern was out of scope of our work.

Reviewer #7: In the current manuscript, Bellini et al. reveal the effectiveness and safety profile of different immunosuppressive regimens in kidney transplantation. A large amount of multicenter data was collected and statistical analysis was performed. Their results indicated that tacrolimus-based immunosuppressive therapy appeared to be superior to cyclosporine in reducing rejection and severe infections. Besides, the combination of tacrolimus and mTORi may represent a valid alternative to the association with mycophenolate. These results are very important to improve graft survival and reduce acute rejection.

1.Mortality and transplant reject/graft failure were defined as outcomes of this study. Is it also possible to include renal function like eGFR as the outcomes? To use renal function as an indicator of effectiveness could help clinicians to evaluate the prognosis. I recommend a deeper exploration of this outcome if it is possible.

Thank you for your suggestion. Even though we agree that renal function would be interesting to investigate in our cohort, we have not included it as one of the outcomes because the study is based on administrative data and we had limited access to information on renal function. The variable related to GFR resulted in a high rate of missing values (exceeding 80% for some of the years considered). We emphasized this aspect in the discussion section.

2. Please unify the wording, such as tumor and cancer.

Thank you for the suggestion, we have replaced the word tumor with cancer in the text.

3.The study relies on medication dispensation data where dosage is lacking. However, medication dosage is a crucial factor to assess the adverse outcome. I noticed hospital information system and co-payment exemption registry. Is it possible to acquire the dosage information by fee items or claims?

Dosage in administrative data can be obtained using “Defined Daily Dose” (DDD), which is the assumed average maintenance dose per day for a drug used for its main indication in adults. However, it is likely that in the context of transplant the therapeutic doses used in clinical practice will be different from the DDD and will depend on individual patient characteristics. This aspect was already mentioned in the limits, we have further clarified it in the discussion.

4. Method - Given the new onset of diabetes and statin use are ones of the outcome in the study, it’s inappropriate to have the history of diabetes and statin use at baseline as covariates. Please excluded patients with these two factors from the beginning.

We had already excluded patients with history of diabetes and statin use from the analysis; we have rephrased the sentence in the methods section for better clarity (L140-142)

5. Result - Table 1 tells several indicators’ SMD value of comparison pairs between TAC+MMF and TAC+mTORi ≥0.1. The indicators should be included as confounders in Cox models to assess the independent effect of exposures.

Ok. As reported in the method we included in the model adjustment for propensity score value 

6. Result - The 112 (20%) patients of TAC+mTORi group were excluded according to the data after propensity score matching, which may result in sampling bias. Please use one to more matching or provide the outcome of the excluded data.

The study’s objective was to examine the relationship between therapies and the outcomes consideres; we opted for statistical methods that prioritize internal validity, albeit potentially reducing external validity.

Rubin DB and colleagues demonstrated that propensity score matching is the best way to eliminate differences between groups (42).

• Rosenbaum PR, Rubin DB. The central role of the propensity score in observational studies for causal effects. Biometrika. 1983;70:41–55.

• Rubin DB. Using multivariate matched sampling and regression adjustment to control bias in observational studies. J Am Stat Assoc. 1979;74:318–328. [Google Scholar]

• Rubin DB, Thomas N. Matching using estimated propensity scores: relating theory to practice. Biometrics. 1996;52:249–264. [PubMed] [Google Scholar]

We added a reference in the paper.

7. Graphics - Figure 2A and 2B were seen unclear here. Please provide clearer figures.

OK. We modified all figures following pacev2

Reviewer #8: (No Response)

---

## [Decision Letter · Decision Letter 3]

16 Nov 2023

Effectiveness and safety of immunosuppressive regimens used as maintenance therapy in kidney transplantation: the CESIT study.

PONE-D-23-20920R3

Dear Dr. Belleudi,

We’re pleased to inform you that your manuscript has been judged scientifically suitable for publication and will be formally accepted for publication once it meets all outstanding technical requirements.

Kind regards,

Yavuz - Ayar

Academic Editor

PLOS ONE

Additional Editor Comments (optional):

Dear Author/s

Greetings

The article can be published in its current form.

Best regards

Reviewers' comments:

Reviewer's Responses to Questions

**Comments to the Author**

1. If the authors have adequately addressed your comments raised in a previous round of review and you feel that this manuscript is now acceptable for publication, you may indicate that here to bypass the “Comments to the Author” section, enter your conflict of interest statement in the “Confidential to Editor” section, and submit your "Accept" recommendation.

Reviewer #7: All comments have been addressed

Reviewer #8: All comments have been addressed

2. Is the manuscript technically sound, and do the data support the conclusions?

Reviewer #7: Yes

Reviewer #8: Yes

3. Has the statistical analysis been performed appropriately and rigorously? 

Reviewer #7: Yes

Reviewer #8: Yes

4. Have the authors made all data underlying the findings in their manuscript fully available?

Reviewer #7: No

Reviewer #8: Yes

5. Is the manuscript presented in an intelligible fashion and written in standard English?

Reviewer #7: Yes

Reviewer #8: Yes

6. Review Comments to the Author

Reviewer #7: (No Response)

Reviewer #8: (No Response)

7. PLOS authors have the option to publish the peer review history of their article (what does this mean?). If published, this will include your full peer review and any attached files.

Reviewer #7: No

Reviewer #8: No

---

## [Editor Report · Acceptance letter]

20 Dec 2023

PONE-D-23-20920R3 

PLOS ONE

Dear Dr. Belleudi, 

I'm pleased to inform you that your manuscript has been deemed suitable for publication in PLOS ONE. Congratulations! Your manuscript is now being handed over to our production team.

Kind regards, 

on behalf of

Professor Yavuz - Ayar 

Academic Editor

PLOS ONE